# Hinge Regression Tree: A Newton Method for Oblique Regression Tree Splitting

**Hongyi Li, Han Lin & Jun Xu**[*†]
School of Intelligence Science and Engineering
Harbin Institute of Technology, Shenzhen
Shenzhen, 518055, China
`{23b904015, 24s153081}@stu.hit.edu.cn`
`xujunqgy@hit.edu.cn`

Code: `https://github.com/Hongyi-Li-sz/Hinge-Regression-Tree`

## Abstract

Oblique decision trees combine the transparency of trees with the power of multivariate decision boundaries—but learning high-quality oblique splits is NP-hard, and practical methods still rely on slow search or theory-free heuristics. We present the Hinge Regression Tree (HRT), which reframes each split as a nonlinear least-squares problem over two linear predictors whose max/min envelope induces ReLU-like expressive power. The resulting alternating fitting procedure is exactly equivalent to a damped Newton (Gauss–Newton) method within fixed partitions. We analyze this node-level optimization and, for a backtracking linesearch variant, prove that the local objective decreases monotonically and converges; in practice, both fixed and adaptive damping yield fast, stable convergence and can be combined with optional ridge regularization. We further prove that HRT's model class is a universal approximator with an explicit $O(\delta^2)$ approximation rate, and show on synthetic and real-world benchmarks that it matches or outperforms single-tree baselines with more compact structures.

## 1 Introduction

Decision trees are among the most influential models in supervised learning due to their interpretability and ability to capture nonlinear relationships. The classical CART framework (Breiman et al., 1984) introduced axis-aligned recursive partitioning, which remains a cornerstone of modern tree-based methods. However, such axis-aligned trees often require deep structures to approximate even simple relationships in high-dimensional or correlated settings, limiting efficiency and generalization (Hastie et al., 2009).

To address these issues, oblique regression trees extend splitting criteria from axis-aligned thresholds to hyperplanes defined by linear combinations of features. This formulation yields more compact structures and improved predictive performance, particularly when features are correlated or rotated (Murthy et al., 1994). Nonetheless, finding the optimal oblique hyperplane is NP-hard (Laurent & Rivest, 1976; Hancock et al., 1996), and practical algorithms have relied on greedy heuristics, evolutionary methods, or convex surrogates (Loh, 2014). Recent work explores optimization-based formulations (Panda et al., 2024; Karthikeyan et al., 2022), but efficient and theoretically sound solutions remain limited.

We introduce a novel oblique regression tree algorithm, termed the Hinge Regression Tree (HRT), that fundamentally redefines the node splitting problem. Specifically, we propose to learn each node split as a nonlinear least squares optimization problem involving two distinct linear models. The basis function employs a hinge formulation to model this nonlinear optimization, intrinsically endowing HRT with ReLU-like nonlinear expressive power. To improve robustness under multicollinearity, ridge regularization (Hoerl & Kennard, 1970) is optionally incorporated within the

---

[*]Also with the Shenzhen Key Lab for Advanced Motion Control and Modern Automation Equipments.
[†]Corresponding author: Jun Xu.

fitting step. The resulting iterative optimization can be interpreted as a damped Newton (Gauss–Newton) method within fixed partitions. Furthermore, we show that, for a backtracking line-search variant, the node-level objective decreases monotonically and converges, and that, when the partition stabilizes, the iterates converge to the corresponding OLS minimizer, while in practice both fixed and adaptive damping exhibit fast and stable convergence. Separately, we establish that the induced piecewise linear model class enjoys strong universal approximation guarantees.

Our contributions can be summarized as follows: (1) We introduce a novel HRT algorithm that reformulates node splitting as a nonlinear least squares optimization over two linear functions. The resulting HRT model, by hierarchically composing its max/min envelope-based splits, intrinsically gains ReLU-like nonlinear expressive power and supports optional ridge regularization. (2) We characterize the node-level alternating optimization as a damped Newton/Gauss–Newton method and, for a backtracking line-search variant, prove that the node-level objective decreases monotonically and converges, thereby providing a theoretical foundation for its efficiency and stable behaviour in practice. (3) We establish that the resulting piecewise linear models constitute a universal approximator with an explicit $O(\delta^2)$ approximation rate, validating HRT's powerful function approximation capabilities. (4) Through extensive experiments on synthetic and real-world datasets, we demonstrate that HRT achieves competitive performance compared to single-tree baselines while maintaining a more compact structure. By integrating regression modeling objectives with optimization theory, our work advances oblique regression trees as both a practical and theoretically principled tool for nonlinear function approximation.

## 2 RELATED WORK

**Oblique regression trees.** The quest for optimal oblique splits dates back decades, driven by their ability to form more compact and expressive decision boundaries compared to axis-aligned trees (Murthy et al., 1994; Loh, 2014). Early approaches largely relied on greedy heuristics, such as OC1 (Murthy et al., 1994) which employed randomized search, or statistical tests for feature selection, like GUIDE (Loh, 2009). However, the NP-hard nature of finding optimal oblique hyperplanes (Laurent & Rivest, 1976) has motivated a shift towards more sophisticated optimization-based methodologies. These include alternating optimization strategies, which iteratively refine split parameters and data assignments (e.g., TAO (Carreira-Perpinán & Tavallali, 2018)), and gradient-based optimization techniques. Recent differentiable oblique trees, such as DGT (Karthikeyan et al., 2022), train trees end-to-end using gradient descent with approximations like over-parameterization and straight-through estimators. Similarly, DTSemNet (Panda et al., 2024) formulates oblique decision trees as neural networks, enabling optimization with standard gradient descent. While these methods have significantly advanced the field, they often rely on heuristics, approximations, or specific neural network architectures. Our work distinguishes itself by reframing the core splitting problem as a direct, nonlinear least squares optimization with a clear equivalence to a damped Newton method, providing a theoretically grounded and efficient alternative.

**Piecewise linear models and hinge functions.** Regression trees are essentially a piecewise linear approximation that recursively partitions the input space and fits a simple model (typically constant or linear) within each region. The universal approximation property of piecewise linear functions is a cornerstone of approximation theory, providing a theoretical foundation for the expressive power of such models (Cybenko, 1989; Hornik, 1991). More recent developments confirm explicit convergence rates for oblique trees (Cattaneo et al., 2024), while deep networks provide complementary insights into piecewise linear expressivity (Hu, 2021). Hinge functions are indispensable piecewise linear primitives in machine learning models, valued for their non-smoothness and geometric interpretability (Ergen & Pilanci, 2021). As a pioneering work, hinging hyperplanes (Breiman, 1993) construct global additive expansions of hinge units. Their influence extends broadly, from inspiring the hinge loss used in SVMs (Cortes & Vapnik, 1995) to motivating the ReLU activation functions that dominate modern neural networks (Nair & Hinton, 2010; Glorot et al., 2011). Owing to their piecewise linear structure, hinge functions are particularly effective for defining efficient and interpretable decision boundaries. However, the reliance on additive combinations in hinging hyperplanes can obscure the interpretability of the resulting models. In contrast, our method directly leverages the characteristics of hinging hyperplanes to formulate node splits, while improving interpretability through explicit expressions of the corresponding subregions.

# 3 TREE CONSTRUCTION AND OBLIQUE SPLIT OPTIMIZATION

This section describes our procedure for constructing the HRT, an oblique decision tree. We first specify the construction objective, then detail how oblique splits are optimized at each internal node, followed by the recursive routine used to construct the full tree. Our key insight is to formulate each split as a nonlinear least-squares problem over two linear models. This formulation allows the tree to directly learn a locally adaptive piecewise-linear structure, from which the splitting hyperplane arises naturally.

## 3.1 CONSTRUCTION OBJECTIVE OF THE HRT

Consider a dataset $\mathcal{S} = \{(\boldsymbol{x}_j, y_j)\}_{j=1}^N$, where $\boldsymbol{x}_j \in \mathbb{R}^d$ and $y_j \in \mathbb{R}$, suppose $\tilde{\boldsymbol{x}} \in \mathbb{R}^{d+1}$ is the augmented feature vector, i.e., $\tilde{\boldsymbol{x}} = [x_1, \dots, x_d, 1]^T$. Assume there are $T$ nodes $\mathbb{D}_t$, indexed in a breadth-first order by $t \in \mathbb{T} = \{1, \dots, T\}$. The nodes can be classified into two types: internal nodes, which execute branching tests and are denoted by $t \in \mathbb{T}_\mathrm{I}$, and leaf nodes, which are utilized for prediction and are denoted by $t \in \mathbb{T}_\mathrm{L}$. Starting from the root node, the tree grows through the internal nodes to reach the leaf nodes. For each node $\mathbb{D}_t$, $t \in \mathbb{T}$, there is an associated parameter vector $\boldsymbol{\theta}_t \in \mathbb{R}^{d+1}$ describing the corresponding linear relationship. Let $t_l(\boldsymbol{x})$ be the leaf index reached by input $\boldsymbol{x}$, and let its linear predictor be $\ell_{t_l}(\boldsymbol{x}) = \tilde{\boldsymbol{x}}^T \boldsymbol{\theta}_{t_l(\boldsymbol{x})}$. The tree's output function is thus $\hat{y}(\boldsymbol{x}) = \ell_{t_l(\boldsymbol{x})}(\boldsymbol{x})$. The global approximation accuracy refers to the training set is then described as

$$J(\mathbb{D}_t, \boldsymbol{\theta}) = \frac{1}{2} \sum_{i=1}^N (y_i - \hat{y}(\boldsymbol{x}_i))^2, \tag{1}$$

where $\hat{y}(\boldsymbol{x}_i) = \ell_{t_l(\boldsymbol{x}_i)}(\boldsymbol{x}_i)$, meaning that the input $\boldsymbol{x}_i$ belongs to a unique leaf node $\mathbb{D}_{t_l(\boldsymbol{x}_i)}$ with $t_l(\boldsymbol{x}_i) \in \mathbb{T}_\mathrm{L}$. The indices of the internal nodes and leaf nodes, i.e., $\mathbb{T}_\mathrm{I}$ and $\mathbb{T}_\mathrm{L}$ are dynamically changing during the construction the HRT.

## 3.2 OPTIMIZATION OF OBLIQUE SPLITS

### 3.2.1 OBJECTIVE FUNCTION

At any internal node $\mathbb{D}_t, t \in \mathbb{T}_\mathrm{I}$, our goal is to find two sets of parameters, $\boldsymbol{\theta}_{t_1}, \boldsymbol{\theta}_{t_2} \in \mathbb{R}^{d+1}$, and these parameters define two linear functions, $\ell_{t_1}(\boldsymbol{x}) = \tilde{\boldsymbol{x}}^T \boldsymbol{\theta}_{t_1}$ and $\ell_{t_2}(\boldsymbol{x}) = \tilde{\boldsymbol{x}}^T \boldsymbol{\theta}_{t_2}$ that best fit the data. Specifically, we aim to minimize the following nonlinear least squares objective function:

$$V(\boldsymbol{\theta}) = \frac{1}{2} \sum_{\boldsymbol{x}_j \in \mathbb{D}_t} (y_j - h(\boldsymbol{x}_j, \boldsymbol{\theta}))^2 \tag{2}$$

with respect to $\boldsymbol{\theta} = [\boldsymbol{\theta}_{t_1}^T, \boldsymbol{\theta}_{t_2}^T]^T \in \mathbb{R}^{2(d+1)}$, which is the total parameter vector, and $h(\boldsymbol{x}_j, \boldsymbol{\theta})$ is defined as:

$$h(\boldsymbol{x}_j, \boldsymbol{\theta}) = \max\left(\tilde{\boldsymbol{x}}_j^T \boldsymbol{\theta}_{t_1}, \tilde{\boldsymbol{x}}_j^T \boldsymbol{\theta}_{t_2}\right) \quad \text{or} \quad h(\boldsymbol{x}_j, \boldsymbol{\theta}) = \min\left(\tilde{\boldsymbol{x}}_j^T \boldsymbol{\theta}_{t_1}, \tilde{\boldsymbol{x}}_j^T \boldsymbol{\theta}_{t_2}\right)$$

The basis function $h(\boldsymbol{x}_j, \boldsymbol{\theta})$ is a hinge function. This function intrinsically defines a decision boundary in the data space, given by the hyperplane $\tilde{\boldsymbol{x}}^T(\boldsymbol{\theta}_{t_1} - \boldsymbol{\theta}_{t_2}) = 0$, where $\ell_{t_1}(\boldsymbol{x}) = \ell_{t_2}(\boldsymbol{x})$. Depending on which side of this hyperplane a data point $\boldsymbol{x}_j$ lies (i.e., whether $\tilde{\boldsymbol{x}}_j^T \boldsymbol{\theta}_{t_1} \geq \tilde{\boldsymbol{x}}_j^T \boldsymbol{\theta}_{t_2}$ or vice versa), the model selects either $\ell_{t_1}(\boldsymbol{x}_j)$ or $\ell_{t_2}(\boldsymbol{x}_j)$ for prediction. This hinge-enabled piecewise linear structure flexibly adapts to both convex and concave local data structures.

From a tree model perspective, minimizing $V(\boldsymbol{\theta})$ corresponds to optimizing a structure with one root and two leaves. The root uses the hyperplane $\tilde{\boldsymbol{x}}^T(\boldsymbol{\theta}_{t_1} - \boldsymbol{\theta}_{t_2}) = 0$ as its decision boundary to partition the data, i.e., the internal node $\mathbb{D}_t$ is partitioned into two leaf nodes $\mathbb{D}_{t_1}$ and $\mathbb{D}_{t_2}$. Each leaf corresponds to a linear function $\ell_{t_1}(\boldsymbol{x})$ or $\ell_{t_2}(\boldsymbol{x})$ used for prediction in their respective data subsets.

### 3.2.2 OPTIMIZATION AS A NEWTON METHOD

We address the nonlinear least squares optimization problem by employing an iterative procedure, which we formulate as a Newton method to derive an efficient solution.

Directly minimizing Equation (2) is difficult due to the non-differentiable function. Our iterative algorithm operates by defining two dynamic partitions of the data based on the current parameters $\boldsymbol{\theta}$:

$$\mathcal{S}_1(\boldsymbol{\theta}) = \{\boldsymbol{x}_j \in \mathbb{D}_t \mid \tilde{\boldsymbol{x}}_j^T \boldsymbol{\theta}_{t_1} \geq \tilde{\boldsymbol{x}}_j^T \boldsymbol{\theta}_{t_2}\}$$
$$\mathcal{S}_2(\boldsymbol{\theta}) = \mathbb{D}_t \setminus \mathcal{S}_1(\boldsymbol{\theta})$$

For a fixed partition $(\mathcal{S}_1, \mathcal{S}_2)$, the objective function is differentiable with respect to $\boldsymbol{\theta}_{t_1}$ and $\boldsymbol{\theta}_{t_2}$ within each respective domain. Under this condition, the Newton update relies on the gradient $\nabla V$ and the Hessian matrix $\nabla^2 V$. Generally, a Newton update is given by $\boldsymbol{\theta}^{(k+1)} = \boldsymbol{\theta}^{(k)} - \mu[\nabla^2 V]^{-1}\nabla V$. In our case, due to the locally linear nature of $h(\boldsymbol{x}_j, \boldsymbol{\theta})$ within each fixed partition, its second derivative is zero, which means the Gauss-Newton approximation to the Hessian is exact. Combining this exact Hessian with the gradient structure (as derived in detail in Appendix A), this update simplifies to:

$$\boldsymbol{\theta}^{(k+1)} = \boldsymbol{\theta}^{(k)} + \mu(\boldsymbol{\theta}_{\text{OLS}}^{(k)} - \boldsymbol{\theta}^{(k)}) \tag{3}$$

where $\boldsymbol{\theta}^{(k)}$ is the current parameter, and $\boldsymbol{\theta}_{\text{OLS}}^{(k)}$ represents the optimal Ordinary Least Squares (OLS) solution independently computed for its respective data subset based on the current partitions $\mathcal{S}_1(\boldsymbol{\theta}^{(k)})$ and $\mathcal{S}_2(\boldsymbol{\theta}^{(k)})$. The term $(\boldsymbol{\theta}_{\text{OLS}}^{(k)} - \boldsymbol{\theta}^{(k)})$ corresponds to the Newton direction $-[\nabla^2 V]^{-1}\nabla V$ for the current fixed partition. $\mu$ is the step size. After $\boldsymbol{\theta}^{(k+1)}$ is computed, the data points are then reassigned to new partitions $\mathcal{S}_1(\boldsymbol{\theta}^{(k+1)})$ and $\mathcal{S}_2(\boldsymbol{\theta}^{(k+1)})$ for the next iteration, based on these updated parameters. This alternating procedure of parameter fitting and partition re-assignment forms the core of our optimization. Our iterative procedure employs a step size $\mu \in (0, 1]$. When $\mu = 1$, the new parameters are directly set to the optimal OLS solution for the current partitions, i.e., $\boldsymbol{\theta}^{(k+1)} = \boldsymbol{\theta}_{\text{OLS}}^{(k)}$, which represents a unit-step Newton update. For a detailed derivation of the general case and its equivalence to OLS when $\mu = 1$, please refer to Appendix A. We support two step-size strategies in practice, both sharing the same Newton direction but differing in how $\mu^{(k)}$ is chosen: (i) a *fixed* damping factor $\mu \in (0, 1]$, which yields a simple and efficient implementation, and (ii) an *automatic* backtracking line search (denoted as "auto" in our experiments), which selects $\mu^{(k)}$ adaptively to ensure sufficient decrease of the node-level objective.

### 3.3 RECURSIVE TREE CONSTRUCTION

The core of the HRT construction is a recursive splitting process. Starting from the root node, each internal node undergoes the node-level optimization procedure described in Section 3.2.2. This procedure iteratively finds the optimal oblique hyperplane, thus creating a linear decision boundary. Data points are subsequently assigned to the left or right child node depending on whether $\tilde{\boldsymbol{x}}^T \boldsymbol{\theta}_{t_1} \geq \tilde{\boldsymbol{x}}^T \boldsymbol{\theta}_{t_2}$. The split is recursively applied to the resulting child nodes until a stopping condition is satisfied, such as reaching the maximum tree depth, falling below a minimum number of samples, or achieving a Root Mean Squared Error (RMSE) below a predefined threshold. Through this recursive mechanism, the tree naturally composes a hierarchy of hinge operations. This hierarchical structure endows the model with ReLU-like expressivity.

To enhance model robustness, all OLS fitting steps can optionally incorporate ridge regression (L2 regularization). Furthermore, to ensure continuous tree growth, even if the node-level iterative optimization fails to converge within a specified number of iterations, a simple fallback mechanism is employed to determine the split. Detailed information on these mechanisms, including the mathematical formulation of ridge regression and the specific implementation of the fallback strategy, can be found in Appendix C. The detailed pseudocode for the optimal node-splitting mechanism and the overall tree-building procedure, along with their computational complexity analysis, can be found in Appendix D and E, respectively.

### 3.4 ILLUSTRATION OF THE RELU-LIKE EXPRESSIVE POWER

Each internal node $\mathbb{D}_t$ carries two linear scores $\ell_{t_1}(\boldsymbol{x}) = \tilde{\boldsymbol{x}}^T \boldsymbol{\theta}_{t_1}$ and $\ell_{t_2}(\boldsymbol{x}) = \tilde{\boldsymbol{x}}^T \boldsymbol{\theta}_{t_2}$. It routes data points by the hinge test $s_t(\boldsymbol{x}) = \ell_{t_1}(\boldsymbol{x}) - \ell_{t_2}(\boldsymbol{x}) \geq 0$ (left child) or $< 0$ (right child). A depth-$k$ tree induces a polyhedral partition of the input space by intersecting halfspaces along root-to-leaf paths. According to Section 3.1, for each input $\boldsymbol{x}$, the tree's output function is $\hat{y}(\boldsymbol{x}) = \ell_{t_l(\boldsymbol{x})}(\boldsymbol{x})$, which

is a piecewise linear function defined on at most $2^k$ regions. Define the ReLU function $\sigma(u) = \max\{u, 0\}$. For any scalars $a, b$, we have $\max(a, b) = a + \sigma(b - a)$ and $\min(a, b) = a - \sigma(a - b)$. This implies that a binary hinge operation at a node is essentially a single-unit ReLU gate acting on linear arguments. Composing these gates along the tree structure yields a computational circuit formed by linear maps and ReLU operations, which justifies its ReLU-like expressive power.

# 4 THEORETICAL ANALYSIS

## 4.1 CONVERGENCE ANALYSIS

Our algorithm uses an alternating optimization strategy. We update parameters by

$$\boldsymbol{\theta}^{(k+1)} = \boldsymbol{\theta}^{(k)} + \mu^{(k)} p^{(k)}, \qquad p^{(k)} = \boldsymbol{\theta}_{\text{OLS}}^{(k)} - \boldsymbol{\theta}^{(k)},$$

where, as shown in Section 3.2, $p^{(k)}$ is a Newton/Gauss–Newton direction under the current partition.

In practice we instantiate this generic update rule in two ways. The first uses a *fixed* damping factor $\mu^{(k)} \equiv \mu \in (0, 1]$; the value of $\mu$ (e.g., 0.1, 0.5, 1) is treated as a hyperparameter and selected on a validation split. The second, denoted as *auto* in our experiments, employs a simple monotone backtracking line search: starting from $\mu^{(k)} = 1$ we geometrically decrease $\mu^{(k)}$ until a valid split with a strictly smaller node-level RMSE is found, or terminate if no such step exists.

For our theoretical analysis we focus on this line-search variant. In Appendix B we show that, under mild assumptions, the resulting damped Newton method yields a monotonically decreasing sequence of node-level objectives that converges, and that when the partition stabilizes the iterates converge to the corresponding OLS minimizer. Empirically, we observe that both the fixed-step and auto line-search schemes exhibit fast and stable convergence. Practical safeguards include optional ridge regularization in the OLS fits to improve conditioning, minimum-sample and error-based stopping at nodes, and a simple fallback (median split) if the node-level optimization fails to make progress within $T_{\max}$ iterations. We terminate when the relative change in $V$ or in $\boldsymbol{\theta}$ falls below a tolerance, or when the partition stabilizes.

## 4.2 UNIVERSAL APPROXIMATION THEORY

This section establishes the theoretical foundation for HRT's expressive power. We prove that its piecewise linear model class is a universal approximator for continuous functions and derive an explicit approximation rate that quantifies how its precision scales with domain partitioning.

**Theorem 1.** *Let $\mathcal{F}$ be the class of piecewise linear functions represented by finite oblique regression trees with linear models at the leaves. Let $g : \mathcal{K} \to \mathbb{R}$ be a twice continuously differentiable function ($g \in C^2(\mathcal{K})$) on a compact set $\mathcal{K} \subset \mathbb{R}^d$. Furthermore, assume that for any constructed partition of $\mathcal{K}$ into regions $\{\mathcal{R}_i\}$ with diameter $\leq \delta$, each region $\mathcal{R}_i$ contains $N_i$ training data points $(\boldsymbol{x}_j, g(\boldsymbol{x}_j))_{j=1}^{N_i}$. Let $X_i$ be the $N_i \times (d+1)$ design matrix whose $j$-th row is $\tilde{\boldsymbol{x}}_j^T$. We further assume that $X_i$ satisfies $\lambda_{\min}(X_i^T X_i) \geq c N_i$ for some constant $c > 0$. Also, $\sup_{\boldsymbol{x} \in \mathcal{K}} \|\tilde{\boldsymbol{x}}\|_2 \leq D_{\mathcal{K}}$ for some bounded constant $D_{\mathcal{K}}$. Then, there exists a constant $C$ such that for any $\delta > 0$, we can construct a function $f \in \mathcal{F}$ by partitioning the domain $\mathcal{K}$ into regions $\{\mathcal{R}_i\}$ with $\max_i \operatorname{diam}(\mathcal{R}_i) \leq \delta$, satisfying the uniform error bound:*

$$\sup_{\boldsymbol{x} \in \mathcal{K}} |f(\boldsymbol{x}) - g(\boldsymbol{x})| \leq C\delta^2$$

*This approximation rate directly implies the universal approximation property, i.e., for any $\epsilon > 0$, there exists a function $f \in \mathcal{F}$ such that $\sup_{\boldsymbol{x} \in \mathcal{K}} |f(\boldsymbol{x}) - g(\boldsymbol{x})| < \epsilon$.*

For the detailed proof, please refer to Appendix F.

**Intuition.** Our HRTs serve as universal approximators by recursively partitioning the input space into convex regions and fitting local linear models in each. As partitions refine (region diameter $\delta$ decreases), the piecewise linear predictor uniformly approximates $g(\boldsymbol{x})$, achieving an $O(\delta^2)$ approximation rate under standard smoothness assumptions. We assume an eigenvalue lower bound

for each leaf $i$, $\lambda_{\min}(X_i^T X_i) \geq c\, N_i$, which keeps local OLS well posed so that its estimation error matches the $O(\delta^2)$ approximation term. In practice, strong collinearity or small leaf sizes may violate this condition; we therefore optionally employ ridge regularization, replacing the local closed form with $(X_i^T X_i + \alpha I_0)^{-1} X_i^T y$. For any $\alpha > 0$, this raises the smallest eigenvalue in the penalized subspace and stabilizes the normal equations.

## 5 EXPERIMENTS

All experiments were conducted using Python 3.11.7 on a machine equipped with an Intel® Xeon® Gold 6530 CPU.

To validate our algorithm's effectiveness and properties, we conducted experiments addressing three key questions: (1) Does the core splitting algorithm converge efficiently as predicted? (2) Can the resulting tree model effectively approximate complex continuous functions, verifying our universal approximation claim? (3) Does our method perform competitively on real-world regression tasks?

### 5.1 CONVERGENCE ANALYSIS OF THE SPLITTING ALGORITHM

**Objective:** This experiment investigates the convergence dynamics of our splitting algorithm, focusing on the critical role of the step size $\mu$. We aim to empirically demonstrate the trade-off between convergence speed and stability by testing the algorithm on two distinct synthetic datasets: one challenging and unstable, and another well-behaved yet non-trivial. This dual approach validates our theoretical claim that while the algorithm is equivalent to a damped Newton method within fixed partitions, a damped step size ($\mu < 1$) is essential for robustness, whereas a unit step can achieve rapid convergence in stable scenarios.

**Setup:** We generate two datasets, each of $N = 1000$ samples with Gaussian noise ($\epsilon \sim \mathcal{N}(0,1)$). For the unstable case, data are generated from the `sinc` function, $y = -\frac{\sin(5\pi x)}{5\pi x} + 0.025\epsilon$, on $x \in [-1.5, 1.5]$. Its multiple local extrema and sharp oscillations create a challenging optimization landscape. For the stable case, data are generated from a smooth, nonlinear function with a distinct inflection point, $y = \frac{2}{1+e^{-3x}} - 0.8x + 0.025\epsilon$, on $x \in [-3, 3]$. This `twisted_sigmoid` function is an ideal well-behaved target, as it requires a nonlinear fit but lacks the sharp features that cause instability. On both datasets, we test four step sizes: $\mu \in \{1.0, 0.5, 0.1, 0.05\}$.

**Results and Analysis:** The results in Figure 1 (unstable case) and Figure 2 (stable case), show the role of the step size. As shown in Figure 1, the results on the `sinc` function highlight the necessity of damping for stability. The unit Newton step ($\mu = 1.0$) is unstable. Its aggressive updates cause a partition collapse within the first few iterations, making the algorithm revert to a poor global linear fit. The large damped step ($\mu = 0.5$), while avoiding outright collapse, becomes trapped in a limit cycle; the model's parameters and the corresponding data partition oscillate between a small set of states without ever converging to a fixed point. Small steps ($\mu = 0.1, 0.05$) prove robust, converging to a high-quality piecewise linear model that accurately captures the function's complex structure. This clearly demonstrates that for challenging problems, effective damping is not merely a heuristic but a prerequisite for success.

Results on the `twisted sigmoid` function (see Figure 2 for visualization and detailed analysis), illustrate the efficiency benefits of the Newton updates. Here, all step sizes converge to the same solution around the function's inflection point. Crucially, the unit Newton step ($\mu = 1.0$) exhibits the fastest convergence, reaching the optimal solution in just a few iterations. As the step size decreases, the number of iterations required for convergence predictably increases, showcasing the classic speed-stability trade-off.

Taken together, these experiments demonstrate our algorithm's behavior. It functions as a damped Newton method. On complex, unstable problems, a small step size ($\mu < 1$) is essential to ensure robust convergence by preventing partition collapse. On well-behaved problems, a unit step size ($\mu = 1$) leverages the full power of the Newton update, achieving extremely rapid convergence. This confirms the algorithm's theoretical foundation and provides a clear practical guideline: the step size $\mu$ serves as a crucial hyperparameter to balance convergence speed against stability, depending on the nature of the problem. Additional ablation studies on the fallback mechanism are reported in Appendix G.

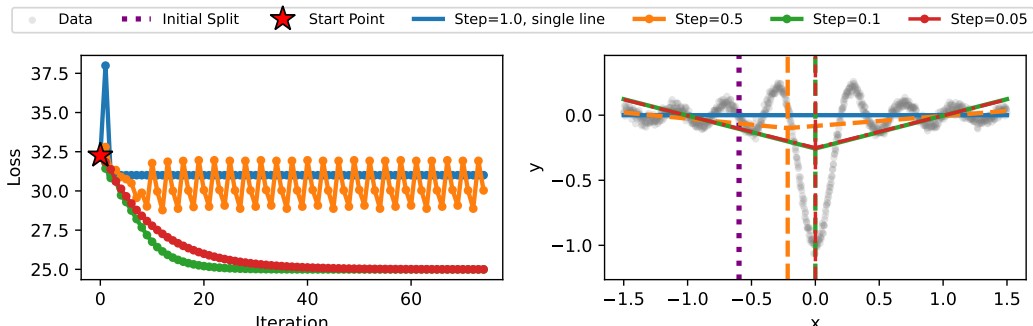

Figure 1: Node-level convergence analysis on the unstable `sinc` function. **Left:** Objective value per iteration for a single internal node (fixed initialization and data subset). The unit step ($\mu = 1.0$, blue) and a large step ($\mu = 0.5$, orange) do not decrease the objective monotonically in this example, and the latter gets trapped in a limit cycle. Smaller damping ($\mu = 0.1$, green; $\mu = 0.05$, red) yields much more regular local Newton dynamics at this node. **Right:** Final fitted models for this controlled experiment. With large steps the node effectively collapses to a poor single linear fit, whereas sufficiently damped updates recover a meaningful piecewise linear approximation. Note that this figure illustrates *local* node-level behaviour; full-tree performance with fallback is analysed in Appendix G.

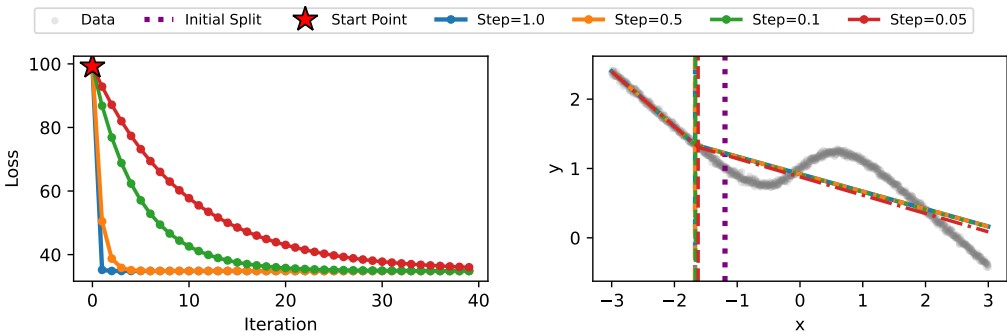

Figure 2: Node-level convergence analysis on the well-behaved `twisted_sigmoid` function. **Left:** For this node, all step sizes lead to monotone decrease of the objective. The unit Newton step ($\mu = 1.0$, blue) reaches the local minimum in the fewest iterations. **Right:** All step sizes arrive at essentially the same high-quality piecewise linear fit around the function's inflection point. This illustrates that, on stable problems, even aggressive Newton steps can behave well at the node level. Global training behaviour across all nodes and datasets is reported in Tables 5–8.

## 5.2 FUNCTION APPROXIMATION ON SYNTHETIC DATA

**Objective:** This experiment validates the piecewise linear approximation of our proposed oblique regression tree and connect it with the theoretical Theorem 1, demonstrating its superiority in fitting complex functions.

**Setup:** We evaluate our method on both 2D and 3D regression tasks. For 2D tasks, we use two classic test functions: the `sinc` function $y = -\frac{\sin(5\pi x)}{5\pi x}$ on $x \in [-1.5, 1.5]$ and the `twisted_sigmoid` function $y = \frac{2}{1+e^{-3x}} - 0.8x$ on $x \in [-3, 3]$. For 3D tasks, we evaluate our method's performance on four oscillatory surface functions with complex characteristics (detailed equations available in Appendix H), with inputs $x_1, x_2$ both ranging from $[-3, 3]$. All synthetic data are generated by adding independent and identically distributed zero-mean Gaussian noise to the true function values. Specifically, the noise standard deviation is $0.025$ for 2D tasks and $0.05$ for 3D tasks. A detailed rationale for the selection of these synthetic functions and noise levels is provided in Appendix H. For each task, we generate a sufficient number of data points (1000 total for 2D tasks, 10000 total for 3D tasks) and split them into 70% for training and 30% for testing.

We compare our piecewise linear regression tree with two representative baseline methods, CART and XGBoost. These baseline models are implemented using the `scikit-learn` library. Our proposed piecewise linear regression tree is trained using Algorithm 4. Hyperparameters for all models are optimized via five-fold cross-validation and grid search on the training set, with final performance reported on the testing set.. For detailed hyperparameter configurations, please refer to Appendix I.

Table 1: Performance comparison on 2D and 3D synthetic functions. All values are reported as mean $\pm$ standard deviation over 10 repetitions for 2D functions and five repetitions for 3D functions. For RMSE and MAE, lower is better ($\downarrow$). For R², higher is better ($\uparrow$). Best results are bolded. Significant improvements over the best baseline are marked with † ($p < 0.05$).

| Model | sinc RMSE($\downarrow$) | MAE($\downarrow$) | R²($\uparrow$) | Model | twisted_sigmoid RMSE($\downarrow$) | MAE($\downarrow$) | R²($\uparrow$) |
|---|---|---|---|---|---|---|---|
| CART | $0.0325 \pm 0.0012$ | $0.0254 \pm 0.0009$ | $0.9831 \pm 0.0049$ | CART | $0.0312 \pm 0.0011$ | $0.0249 \pm 0.0009$ | $0.9976 \pm 0.0002$ |
| XGB | $0.0289 \pm 0.0010$ | $0.0228 \pm 0.0009$ | $0.9868 \pm 0.0031$ | XGB | $0.0286 \pm 0.0005$ | $0.0226 \pm 0.0007$ | $0.9980 \pm 0.0001$ |
| Ours | $\mathbf{0.0280 \pm 0.0009}$ | $\mathbf{0.0222 \pm 0.0008}$ | $\mathbf{0.9876 \pm 0.0028}$ | Ours | $\mathbf{0.0258 \pm 0.0004}$† | $\mathbf{0.0205 \pm 0.0003}$† | $\mathbf{0.9983 \pm 0.0001}$† |
| | $f_1(x_1, x_2)$ | | | | $f_2(x_1, x_2)$ | | |
| CART | $0.8552 \pm 0.0148$ | $0.6580 \pm 0.0127$ | $0.9945 \pm 0.0001$ | CART | $0.2721 \pm 0.0028$ | $0.1972 \pm 0.0024$ | $0.9301 \pm 0.0011$ |
| XGB | $0.3018 \pm 0.0088$ | $0.2317 \pm 0.0069$ | $0.9993 \pm 0.0000$ | XGB | $0.0859 \pm 0.0015$ | $0.0681 \pm 0.0013$ | $0.9930 \pm 0.0003$ |
| Ours | $\mathbf{0.1646 \pm 0.0796}$† | $\mathbf{0.1080 \pm 0.0362}$† | $\mathbf{0.9998 \pm 0.0003}$† | Ours | $\mathbf{0.0757 \pm 0.0020}$† | $\mathbf{0.0587 \pm 0.0016}$† | $\mathbf{0.9946 \pm 0.0003}$† |
| | $f_3(x_1, x_2)$ | | | | $f_4(x_1, x_2)$ | | |
| CART | $0.0752 \pm 0.0011$ | $0.0587 \pm 0.0007$ | $0.9832 \pm 0.0005$ | CART | $0.0789 \pm 0.0014$ | $0.0575 \pm 0.0005$ | $0.9945 \pm 0.0002$ |
| XGB | $0.0537 \pm 0.0003$ | $0.0428 \pm 0.0001$ | $0.9914 \pm 0.0002$ | XGB | $0.0556 \pm 0.0005$ | $0.0438 \pm 0.0003$ | $\mathbf{0.9973 \pm 0.0001}$ |
| Ours | $\mathbf{0.0528 \pm 0.0004}$† | $\mathbf{0.0420 \pm 0.0003}$† | $\mathbf{0.9917 \pm 0.0002}$† | Ours | $\mathbf{0.0555 \pm 0.0007}$ | $\mathbf{0.0436 \pm 0.0004}$ | $\mathbf{0.9973 \pm 0.0001}$ |

**2D and 3D Experimental Results** The results, summarized in Table 1, demonstrate our method's ability to effectively approximate complex functions across both 2D and 3D tasks. For a detailed visual comparison of 2D approximation performance and visualizations for $f_1$, $f_2$, $f_3$ and $f_4$, please refer to Appendix H. The superior performance of our method across various synthetic functions arises from two fundamental design principles. First, the combination of piecewise linear modeling and Newton iterative optimization allows for precise and efficient local approximation. In contrast to traditional piecewise constant tree models, we explicitly fit linear or planar functions within each partition and employ Newton updates to quickly converge to high-quality solutions. Second, flexible oblique splits significantly enhance the adaptability of feature-space partitioning. By overcoming the limitations of axis-parallel splits, they yield decision boundaries that better align with the intrinsic geometry of the data, enabling more efficient and accurate region segmentation.

## 5.3 Performance on Real-World Regression Datasets

**Objective:** To evaluate the practical performance of our proposed model on standard benchmark regression datasets and compare it against other well-established methods.

**Setup:** To assess the practical performance of the proposed method, we evaluate it on a diverse suite of publicly available regression datasets, including all seven benchmarks from Zharmagambetov et al. (2021) and several large-scale industrial datasets such as *YearPred*. These datasets cover a wide spectrum of feature dimensionalities ($N_f$) and sample sizes ($N_s$), from low-dimensional, small-sample tasks to high-dimensional, large-scale scenarios (see Table 2).

We compare against the following strong baselines: DTSemNet (Panda et al., 2024), DGT (Karthikeyan et al., 2022), TAO (oblique and axis-aligned) (Carreira-Perpiñán & Tavallali, 2018; Zharmagambetov et al., 2021), CART (Breiman et al., 1984), M5 model trees (Wang & Witten, 1997), the linear trees implemented in the `linear-tree` library (Cerliani, 2022), and XGBoost (Chen & Guestrin, 2016) (an ensemble method).

For the seven datasets from Zharmagambetov et al. (2021), we directly cite performance from that work. For the remaining datasets without published results, we reproduce results. The only exception is TAO, which is not publicly available; its reported numbers are directly taken from the original paper. For reproduced experiments, hyperparameters are tuned via five-fold cross-validation on the training set, with final performance reported on the testing set. Specifically, for datasets with pre-defined train/test splits, we adhere to these original partitions. For datasets without such predefined splits, we first randomly divide the entire dataset into a 0.5:0.5 train/test ratio. For detailed hyperparameter configurations, please refer to Appendix I.

Table 2: Average RMSE results (lower is better) on regression tasks, ± std over five runs. The first column lists the dataset name, followed by the number of features $N_f$ and the number of samples $N_s$. The first seven datasets are from Zharmagambetov et al. (2021). The reported results for TAO (including TAO-A: axis-aligned and TAO-O: oblique) and CART are taken from Zharmagambetov et al. (2021); DGT results from Karthikeyan et al. (2022); DTSemNet results from Panda et al. (2024); XGBoost results from Zharmagambetov & Carreira-Perpinan (2020). For datasets with reported results in the original papers, we directly cite their results. For datasets without reported results, we reproduce the experiments under the same settings. The only exception is TAO, which is not publicly available. Note that for the Ailerons, D-Elevators, and D-Ailerons datasets, the reported RMSE values are scaled by $\times 10^{-4}$, $\times 10^{-3}$, and $\times 10^{-4}$ respectively. A dash (–) indicates runtime exceeding 10 hours. Significant improvements over the best baseline are marked with † ($p < 0.05$).

| Dataset ($N_f$, $N_s$) | DTSemNet | DGT | TAO-A | TAO-O | CART | XGB | M5 | Linear tree | HRT (ours) |
|---|---|---|---|---|---|---|---|---|---|
| Abalone (8, 4k) | $2.14 \pm 0.03$ | $2.15 \pm 0.03$ | $2.32 \pm 0.58$ | $2.18 \pm 0.05$ | $2.34 \pm 0.59$ | $2.20 \pm 0.00$ | $2.18 \pm 0.04$ | $2.22 \pm 0.05$ | $\mathbf{2.11 \pm 0.05}$ |
| CPUact (21, 8k) | $2.65 \pm 0.18$ | $2.91 \pm 0.15$ | $3.26 \pm 0.51$ | $2.71 \pm 0.04$ | $3.28 \pm 0.44$ | $2.57 \pm 0.00$ | $4.16 \pm 2.72$ | $3.05 \pm 0.66$ | $\mathbf{2.56 \pm 0.05}$ |
| Ailerons (40, 14k) | $1.66 \pm 0.01$ | $1.72 \pm 0.02$ | $2.55 \pm 0.00$ | $1.76 \pm 0.02$ | $2.85 \pm 0.57$ | $1.72 \pm 0.00$ | $1.68 \pm 0.00$ | $1.75 \pm 0.00$ | $\mathbf{1.64 \pm 0.00}$† |
| CTSlice (384, 54k) | $1.45 \pm 0.12$ | $2.30 \pm 0.17$ | $2.66 \pm 0.04$ | $1.54 \pm 0.05$ | $2.69 \pm 0.03$ | $\mathbf{1.18 \pm 0.00}$ | $3.81 \pm 0.28$ | $2.89 \pm 0.67$ | $1.41 \pm 0.15$ |
| YearPred (90, 515k) | $8.99 \pm 0.01$ | $9.05 \pm 0.01$ | $9.76 \pm 0.11$ | $9.11 \pm 0.05$ | $9.79 \pm 0.54$ | $9.01 \pm 0.00$ | $9.43 \pm 0.03$ | $9.15 \pm 0.01$ | $\mathbf{8.97 \pm 0.02}$ |
| Concrete (8, 1k) | $7.96 \pm 0.33$ | $7.43 \pm 0.24$ | $7.20 \pm 3.17$ | $7.17 \pm 0.43$ | $7.22 \pm 3.13$ | $6.75 \pm 0.21$ | $\mathbf{6.65 \pm 0.12}$ | $6.93 \pm 0.24$ | $6.92 \pm 0.24$ |
| Airfoil (5, 2k) | $3.83 \pm 0.16$ | $3.72 \pm 0.10$ | $2.73 \pm 0.62$ | $3.13 \pm 0.38$ | $2.75 \pm 0.62$ | $2.77 \pm 0.10$ | $3.21 \pm 0.06$ | $2.81 \pm 0.12$ | $\mathbf{2.63 \pm 0.10}$ |
| Fried (10, 41k) | $1.51 \pm 0.00$ | $2.27 \pm 0.09$ | N/A | N/A | $1.96 \pm 0.01$ | $\mathbf{1.09 \pm 0.00}$ | $1.59 \pm 0.02$ | $1.10 \pm 0.00$ | $\mathbf{1.09 \pm 0.01}$† |
| D-Elevators (6, 10k) | $1.46 \pm 0.00$ | $1.46 \pm 0.01$ | N/A | N/A | $1.50 \pm 0.02$ | $1.46 \pm 0.02$ | $1.43 \pm 0.02$ | $1.45 \pm 0.02$ | $\mathbf{1.43 \pm 0.01}$ |
| D-Ailerons (5, 7k) | $1.76 \pm 0.00$ | $1.75 \pm 0.02$ | N/A | N/A | $1.83 \pm 0.04$ | $1.72 \pm 0.05$ | $\mathbf{1.65 \pm 0.03}$ | $1.72 \pm 0.02$ | $1.68 \pm 0.02$ |
| Kinematics (8, 8k) | $0.168 \pm 0.000$ | $0.135 \pm 0.003$ | N/A | N/A | $0.200 \pm 0.003$ | $0.125 \pm 0.001$ | $0.175 \pm 0.004$ | $0.141 \pm 0.003$ | $\mathbf{0.102 \pm 0.003}$† |
| C&C (127, 2k) | $0.201 \pm 0.000$ | $0.227 \pm 0.000$ | N/A | N/A | $0.160 \pm 0.005$ | $0.142 \pm 0.004$ | $0.145 \pm 0.004$ | $0.195 \pm 0.009$ | $\mathbf{0.140 \pm 0.004}$ |
| Blog (280, 60k) | $32.30 \pm 0.00$ | $28.35 \pm 1.57$ | N/A | N/A | $26.78 \pm 2.39$ | $23.02 \pm 0.00$ | $22.81 \pm 0.11$ | - | $25.41 \pm 1.83$ |

**Results and Analysis:** As shown in Table 2, our method achieves the best or highly competitive RMSE across a broad set of datasets. Among all single-tree models, our method demonstrates superior performance on the vast majority of datasets. In most datasets, the proposed approach yields notable reductions in RMSE. Even for large-scale datasets like *YearPred*, our performance remains on par with or occasionally superior to the best existing results (including ensemble methods), demonstrating the scalability of our approach. For hyperparameter configurations, please refer to Appendix I.

In addition to RMSE, Table 3 assesses the structural complexity of the learned models in terms of *average depth* ($\Delta$) and *average number of leaves* (L) over five runs. Our method typically produces significantly shallower trees with fewer leaves compared to other single-tree baselines. For instance, on *Concrete*, our model achieves competitive RMSE with a depth of only 3 and 5.8 leaves, demonstrating superior performance compared to CART, which requires a depth of 11.2 and 113 leaves to attain higher error. This indicates that the proposed method attains a favorable trade-off between *predictive performance* and *transparency*. Regarding training time, as shown in Table 4, our method also shows efficient training performance on multiple datasets. While all experiments in this section focus on regression, a preliminary extension of HRT to binary classification is reported in Appendix K, where the same framework achieves competitive AUC and F1 scores with substantially shallower trees.

Table 3: Average depths ($\Delta$) and average number of leaves (L) over five repetitions for regression datasets. Only single-tree models are included in this comparison; XGBoost is excluded, as its depth and leaf statistics are not comparable to those of single decision trees. A dash (–) indicates that the corresponding result was not reported in the original source. Datasets and results for TAO (including TAO-A: axis-aligned and TAO-O: oblique) and CART are from Zharmagambetov et al. (2021), DGT results are from Karthikeyan et al. (2022), and DTSemNet results are from Panda et al. (2024).

| Dataset ($N_f$, $N_s$) | DTSemNet | | DGT | | TAO-A | | TAO-O | | CART | | Ours | |
|---|---|---|---|---|---|---|---|---|---|---|---|---|
| | $\Delta$ | L | $\Delta$ | L | $\Delta$ | L | $\Delta$ | L | $\Delta$ | L | $\Delta$ | L |
| Abalone (8, 4k) | 5.0 | – | 6.0 | – | 5.0 | 12.8 | 6.0 | 58.6 | 5.0 | 12.8 | **2.0** | **4.0** |
| CPUact (21, 8k) | 5.0 | – | 6.0 | – | 9.0 | 57.2 | 6.0 | 52.7 | 9.0 | 57.2 | **2.0** | **4.0** |
| Ailerons (40, 14k) | 5.0 | – | 6.0 | – | 7.0 | 15.0 | 6.0 | 60.2 | 7.0 | 15.0 | **2.0** | **4.0** |
| CTSlice (384, 54k) | 5.0 | – | 10.0 | – | 36.0 | 700.0 | **7.0** | **74.8** | 36.0 | 700.0 | 7.0 | 93.2 |
| YearPred (90, 515k) | 6.0 | – | 8.0 | – | 12.0 | 135.0 | 8.0 | 157.9 | 12.0 | 135.0 | **4.0** | **16.0** |
| Concrete (8, 1k) | 5.0 | – | 6.0 | – | 11.2 | 113.0 | 9.0 | 192.0 | 11.2 | 113.0 | **3.0** | **5.8** |
| Airfoil (5, 2k) | **5.0** | – | 6.0 | – | 15.0 | 479.8 | 8.0 | 147.1 | 15.0 | 479.8 | 5.0 | 25.0 |

Table 4: Training time (in seconds, lower is better) on four regression datasets. The first column lists the dataset name, followed by the number of features $N_f$ and the number of samples $N_s$. The numbers in parentheses represent the average number of iterations for HRT (ours).

| Dataset ($N_f$, $N_s$) | DTSemNet | DGT | CART | XGB | M5 | Linear tree | HRT (ours) |
|---|---|---|---|---|---|---|---|
| D-Ailerons (5, 7k) | 9.557 | 60.850 | 0.008 | 2.757 | 0.344 | 3.601 | 0.183 (6.7) |
| Fried (10, 41k) | 69.979 | 395.243 | 0.171 | 9.722 | 8.322 | 25.743 | 2.807 (45.7) |
| Kinematics (8, 8k) | 16.950 | 93.690 | 0.028 | 14.064 | 1.447 | 7.960 | 0.790 (15.8) |
| C&C (127, 2k) | 0.908 | 51.548 | 0.057 | 53.801 | 0.177 | 307.832 | 0.178 (1.4) |

## 6    CONCLUSION

We presented the Hinge Regression Tree (HRT), a new oblique regression tree method that formulates each split as a nonlinear least-squares fit over two linear models connected by a hinge. This yields a hierarchy of max/min envelope splits that provides ReLU-like piecewise linear expressiveness while preserving the interpretability of shallow decision trees.

We showed that the alternating fitting step at each node corresponds to a damped Newton/Gauss–Newton procedure over fixed partitions. With a backtracking line search, we proved monotonic decrease of the node objective and convergence to the OLS solution once the partition stabilizes. In practice, both fixed and adaptive damping schemes converge quickly and stably, and can incorporate ridge regularization for robustness. From a functional viewpoint, we also established a universal approximation property for the resulting piecewise-linear model class, with an explicit $O(\delta^2)$ rate that links approximation error to partition granularity.

Experiments on synthetic and real datasets demonstrate that HRT matches or exceeds strong single-tree baselines while producing significantly shallower trees with fewer leaves, achieving a favorable balance between accuracy, compactness, and interpretability.

Future directions include extending the method to classification and structured outputs, developing more adaptive step-size and regularization strategies for large-scale or ill-conditioned settings, and integrating HRT-style node optimization into ensemble methods such as boosting and random forests.

## REPRODUCIBILITY STATEMENT

We have taken several steps to ensure the reproducibility of this work. The detailed mathematical derivations for our novel node-splitting algorithm's optimization strategy and its equivalence to a damped Newton method are thoroughly presented in Appendix A. Complete pseudocode for both the optimal node-splitting mechanism (Algorithm 2) and the overall tree-building procedure (Algorithm 4) is provided in Appendix D. Additionally, Appendix C outlines the optional ridge regression regularization and the fallback strategy for non-convergent nodes. The theoretical analysis, including convergence properties (further supported by empirical analysis in Section 5.1) and the proofs for the universal approximation property, are detailed in Appendix A and Appendix F respectively. For the experimental setup, comprehensive descriptions of synthetic data generation and definitions (Appendix H), real-world datasets (Table 13), and hyperparameter configurations for both baseline methods and our approach (Tables 9, 10, 11, and 12) are fully documented in Appendix I.

## ACKNOWLEDGMENTS

This work was supported in part by the Science and Technology Innovation Committee of Shenzhen Municipality under Grant GXWD20231129101652001, in part by the Shenzhen Science and Technology Program under Grant SYSPG20241211173609005, and in part by Guangdong Provincial Key Laboratory of Intelligent Morphing Mechanisms and Adaptive Robotics under Grant 2023B1212010005.

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

# APPENDIX

# A   DETAILED DERIVATIONS FOR EQUIVALENCE OF ITERATIVE OPTIMIZATION TO NEWTON METHOD

This appendix provides a detailed mathematical derivation demonstrating the equivalence of our iterative optimization procedure to a Newton method.

Recall from Section 3 that at any internal node of the tree $\mathbb{D}_t$, for all $\boldsymbol{x}_j \in \mathbb{D}_t$, we aim to minimize the nonlinear least squares objective function:

$$V(\boldsymbol{\theta}) = \frac{1}{2} \sum_{\boldsymbol{x}_j \in \mathbb{D}_t} (y_j - h(\boldsymbol{x}_j, \boldsymbol{\theta}))^2$$

where $\boldsymbol{\theta} = [\boldsymbol{\theta}_{t_1}^T, \boldsymbol{\theta}_{t_2}^T]^T \in \mathbb{R}^{2(d+1)}$ is the total parameter vector. The function $h(\boldsymbol{x}_j, \boldsymbol{\theta})$ is defined as $h(\boldsymbol{x}_j, \boldsymbol{\theta}) = \max\left(\tilde{\boldsymbol{x}}_j^T \boldsymbol{\theta}_{t_1}, \tilde{\boldsymbol{x}}_j^T \boldsymbol{\theta}_{t_2}\right)$ or $h(\boldsymbol{x}_j, \boldsymbol{\theta}) = \min\left(\tilde{\boldsymbol{x}}_j^T \boldsymbol{\theta}_{t_1}, \tilde{\boldsymbol{x}}_j^T \boldsymbol{\theta}_{t_2}\right)$. For clarity, we will provide the detailed derivation for the $\max$ form; the derivation for the $\min$ form follows analogously with adjusted partition definitions. As discussed in Section 3, directly minimizing $V(\boldsymbol{\theta})$ is challenging due to the non-differentiability of the hinge functions at the switching points.

Our approach can be interpreted as an iterative procedure equivalent to the Newton method within fixed partitions. We first define dynamic partitions of the data based on the current parameters $\boldsymbol{\theta}$, as introduced in Section 3:

$$\mathcal{S}_1(\boldsymbol{\theta}) = \{\boldsymbol{x}_j \in \mathbb{D}_t \mid \tilde{\boldsymbol{x}}_j^T \boldsymbol{\theta}_{t_1} \geq \tilde{\boldsymbol{x}}_j^T \boldsymbol{\theta}_{t_2}\}$$
$$\mathcal{S}_2(\boldsymbol{\theta}) = \mathbb{D}_t \setminus \mathcal{S}_1(\boldsymbol{\theta})$$

For a fixed partition $(\mathcal{S}_1, \mathcal{S}_2)$, the objective function is differentiable with respect to $\boldsymbol{\theta}_{t_1}$ and $\boldsymbol{\theta}_{t_2}$ within their respective domains. Under this condition, we can compute the gradient $\nabla V$ and the Hessian matrix $\nabla^2 V$.

**Gradient Calculation.** For almost all $\boldsymbol{\theta}$ (where no data point lies exactly on the boundary $\tilde{\boldsymbol{x}}_j^T \boldsymbol{\theta}_{t_1} = \tilde{\boldsymbol{x}}_j^T \boldsymbol{\theta}_{t_2}$), the model function $h(\boldsymbol{x}_j, \boldsymbol{\theta})$ is differentiable at each data point. Specifically, for $h(\boldsymbol{x}_j, \boldsymbol{\theta}) = \max\left(\tilde{\boldsymbol{x}}_j^T \boldsymbol{\theta}_{t_1}, \tilde{\boldsymbol{x}}_j^T \boldsymbol{\theta}_{t_2}\right)$,

$$\nabla h(\boldsymbol{x}_j, \boldsymbol{\theta}) = \begin{cases} \begin{pmatrix} \tilde{\boldsymbol{x}}_j \\ \mathbf{0} \end{pmatrix} & \text{if } \tilde{\boldsymbol{x}}_j^T \boldsymbol{\theta}_{t_1} > \tilde{\boldsymbol{x}}_j^T \boldsymbol{\theta}_{t_2} \\ \begin{pmatrix} \mathbf{0} \\ \tilde{\boldsymbol{x}}_j \end{pmatrix} & \text{if } \tilde{\boldsymbol{x}}_j^T \boldsymbol{\theta}_{t_1} < \tilde{\boldsymbol{x}}_j^T \boldsymbol{\theta}_{t_2} \end{cases}$$

Thus, the gradient of the objective function is:

$$\nabla V(\boldsymbol{\theta}) = \sum_{j=1}^N (y_j - h(\boldsymbol{x}_j, \boldsymbol{\theta}))(-\nabla h(\boldsymbol{x}_j, \boldsymbol{\theta})) = -\sum_{j=1}^N (y_j - h(\boldsymbol{x}_j, \boldsymbol{\theta}))\nabla h(\boldsymbol{x}_j, \boldsymbol{\theta})$$

Substituting $\nabla h(\boldsymbol{x}_j, \boldsymbol{\theta})$ and summing over the respective partitions:

$$\nabla V(\boldsymbol{\theta}) = -\begin{pmatrix} \sum_{(\boldsymbol{x}_j, y_j) \in \mathcal{S}_1(\boldsymbol{\theta})} \tilde{\boldsymbol{x}}_j(y_j - \tilde{\boldsymbol{x}}_j^T \boldsymbol{\theta}_{t_1}) \\ \sum_{(\boldsymbol{x}_j, y_j) \in \mathcal{S}_2(\boldsymbol{\theta})} \tilde{\boldsymbol{x}}_j(y_j - \tilde{\boldsymbol{x}}_j^T \boldsymbol{\theta}_{t_2}) \end{pmatrix}$$

**Hessian Matrix Calculation.** For a generic nonlinear least squares problem, the Hessian matrix is given by:

$$\nabla^2 V(\boldsymbol{\theta}) = \sum_j [(\nabla h(\boldsymbol{x}_j, \boldsymbol{\theta}))(\nabla h(\boldsymbol{x}_j, \boldsymbol{\theta}))^T - (y_j - h(\boldsymbol{x}_j, \boldsymbol{\theta}))\nabla^2 h(\boldsymbol{x}_j, \boldsymbol{\theta})]$$

However, in our specific case, because $h(\boldsymbol{x}_j, \boldsymbol{\theta})$ is locally linear within each fixed partition (i.e., $h(\boldsymbol{x}_j, \boldsymbol{\theta})$ is either $\tilde{\boldsymbol{x}}_j^T \boldsymbol{\theta}_{t_1}$ or $\tilde{\boldsymbol{x}}_j^T \boldsymbol{\theta}_{t_2}$), its second derivative $\nabla^2 h(\boldsymbol{x}_j, \boldsymbol{\theta})$ is zero. This causes the second term of the Hessian to vanish, meaning the expression $\sum_j (\nabla h_j)(\nabla h_j)^T$, often used as the Gauss-Newton approximation, becomes the exact Hessian.

Therefore, the exact Hessian matrix is:

$$\nabla^2 V(\boldsymbol{\theta}) = \begin{pmatrix} \sum_{(\boldsymbol{x}_j, y_j) \in \mathcal{S}_1(\boldsymbol{\theta})} \tilde{\boldsymbol{x}}_j \tilde{\boldsymbol{x}}_j^T & \mathbf{0} \\ \mathbf{0} & \sum_{(\boldsymbol{x}_j, y_j) \in \mathcal{S}_2(\boldsymbol{\theta})} \tilde{\boldsymbol{x}}_j \tilde{\boldsymbol{x}}_j^T \end{pmatrix}$$

where $\mathbf{0}$ is a $(d+1) \times (d+1)$ zero matrix.

**Newton Update and OLS Equivalence.** The Newton update at iteration $k$ is generally given by $\boldsymbol{\theta}^{(k+1)} = \boldsymbol{\theta}^{(k)} - \mu[\nabla^2 V(\boldsymbol{\theta}^{(k)})]^{-1}\nabla V(\boldsymbol{\theta}^{(k)})$. Let $\boldsymbol{\theta}^{(k)} = [\boldsymbol{\theta}_{t_1}^{(k)T}, \boldsymbol{\theta}_{t_2}^{(k)T}]^T$. We examine the updates for $\boldsymbol{\theta}_{t_1}$ and $\boldsymbol{\theta}_{t_2}$ separately.

For $\boldsymbol{\theta}_{t_1}$, the update is:

$$
\begin{aligned}
\boldsymbol{\theta}_{t_1}^{(k+1)} &= \boldsymbol{\theta}_{t_1}^{(k)} - \mu \left( \sum_{(\boldsymbol{x}_j, y_j) \in \mathcal{S}_1(\boldsymbol{\theta}^{(k)})} \tilde{\boldsymbol{x}}_j \tilde{\boldsymbol{x}}_j^T \right)^{-1} \left( -\sum_{(\boldsymbol{x}_j, y_j) \in \mathcal{S}_1(\boldsymbol{\theta}^{(k)})} \tilde{\boldsymbol{x}}_j (y_j - \tilde{\boldsymbol{x}}_j^T \boldsymbol{\theta}_{t_1}^{(k)}) \right) \\
&= \boldsymbol{\theta}_{t_1}^{(k)} + \mu \left( \sum_{(\boldsymbol{x}_j, y_j) \in \mathcal{S}_1^{(k)}} \tilde{\boldsymbol{x}}_j \tilde{\boldsymbol{x}}_j^T \right)^{-1} \left( \sum_{(\boldsymbol{x}_j, y_j) \in \mathcal{S}_1^{(k)}} \tilde{\boldsymbol{x}}_j y_j - \sum_{(\boldsymbol{x}_j, y_j) \in \mathcal{S}_1^{(k)}} \tilde{\boldsymbol{x}}_j \tilde{\boldsymbol{x}}_j^T \boldsymbol{\theta}_{t_1}^{(k)} \right) \\
&= \boldsymbol{\theta}_{t_1}^{(k)} + \mu \left[ \left( \sum_{(\boldsymbol{x}_j, y_j) \in \mathcal{S}_1^{(k)}} \tilde{\boldsymbol{x}}_j \tilde{\boldsymbol{x}}_j^T \right)^{-1} \left( \sum_{(\boldsymbol{x}_j, y_j) \in \mathcal{S}_1^{(k)}} \tilde{\boldsymbol{x}}_j y_j \right) - \boldsymbol{\theta}_{t_1}^{(k)} \right]
\end{aligned}
$$

Let $\boldsymbol{\theta}_{\mathrm{OLS},1}^{(k)} = \left( \sum_{(\boldsymbol{x}_j, y_j) \in \mathcal{S}_1^{(k)}} \tilde{\boldsymbol{x}}_j \tilde{\boldsymbol{x}}_j^T \right)^{-1} \left( \sum_{(\boldsymbol{x}_j, y_j) \in \mathcal{S}_1^{(k)}} \tilde{\boldsymbol{x}}_j y_j \right)$ be the OLS solution for the data subset $\mathcal{S}_1^{(k)}$. Then the Newton update can be written as:

$$
\boldsymbol{\theta}_{t_1}^{(k+1)} = \boldsymbol{\theta}_{t_1}^{(k)} + \mu(\boldsymbol{\theta}_{\mathrm{OLS},1}^{(k)} - \boldsymbol{\theta}_{t_1}^{(k)})
$$

A similar derivation holds for $\boldsymbol{\theta}_2$:

$$
\boldsymbol{\theta}_{t_2}^{(k+1)} = \boldsymbol{\theta}_{t_2}^{(k)} + \mu(\boldsymbol{\theta}_{\mathrm{OLS},2}^{(k)} - \boldsymbol{\theta}_{t_2}^{(k)})
$$

where $\boldsymbol{\theta}_{\mathrm{OLS},2}^{(k)}$ is the OLS solution for the data subset $\mathcal{S}_2^{(k)}$.

In our algorithm, the step size $\mu$ is a tunable parameter. When a unit step size is employed (i.e., $\mu = 1$), the update formulas simplify to:

$$
\boldsymbol{\theta}_{t_1}^{(k+1)} = \boldsymbol{\theta}_{\mathrm{OLS},1}^{(k)}
$$

$$
\boldsymbol{\theta}_{t_2}^{(k+1)} = \boldsymbol{\theta}_{\mathrm{OLS},2}^{(k)}
$$

This demonstrates that when a unit step size ($\mu = 1$) is employed, the alternating procedure of model fitting (performing OLS on the current partitions) and partitioning (reassigning data points based on the new model parameters), as presented in Algorithm 2, is precisely equivalent to executing a unit-step Newton method to minimize the original nonlinear objective function $V(\boldsymbol{\theta})$. This choice of $\mu = 1$ is common for this type of problem due to its direct connection to the optimal OLS solutions within fixed partitions.

## B  NODE-LEVEL CONVERGENCE OF THE LINE-SEARCH NEWTON UPDATE

This appendix provides a node-level convergence guarantee for the line-search Newton update described in Section 3.2.2. The analysis is local to a single internal node and does not rely on global tree construction. Throughout, we adopt the notation introduced in Section 3.2.1.

**Setup.** At an internal node $\mathbb{D}_t$, the local objective is

$$
V(\boldsymbol{\theta}) = \frac{1}{2} \sum_{\boldsymbol{x}_j \in \mathbb{D}_t} \left( y_j - h(\boldsymbol{x}_j, \boldsymbol{\theta}) \right)^2, \qquad \boldsymbol{\theta} = [\boldsymbol{\theta}_{t_1}^T, \boldsymbol{\theta}_{t_2}^T]^T \in \mathbb{R}^{2(d+1)}, \tag{4}
$$

where

$$
h(\boldsymbol{x}_j, \boldsymbol{\theta}) = \max\{\tilde{\boldsymbol{x}}_j^T \boldsymbol{\theta}_{t_1}, \tilde{\boldsymbol{x}}_j^T \boldsymbol{\theta}_{t_2}\}, \qquad \tilde{\boldsymbol{x}}_j = [\boldsymbol{x}_j^T, 1]^T.
$$

Given $\boldsymbol{\theta}$, define the partition

$$\mathcal{S}_1(\boldsymbol{\theta}) = \{\boldsymbol{x}_j \in \mathbb{D}_t : \tilde{\boldsymbol{x}}_j^T \boldsymbol{\theta}_{t_1} \geq \tilde{\boldsymbol{x}}_j^T \boldsymbol{\theta}_{t_2}\}, \tag{5}$$

$$\mathcal{S}_2(\boldsymbol{\theta}) = \mathbb{D}_t \setminus \mathcal{S}_1(\boldsymbol{\theta}). \tag{6}$$

On a fixed partition $(\mathcal{S}_1, \mathcal{S}_2)$, the hinge becomes inactive and

$$V(\boldsymbol{\theta}) = \tfrac{1}{2}\|\boldsymbol{y}_1 - X_1\boldsymbol{\theta}_{t_1}\|_2^2 + \tfrac{1}{2}\|\boldsymbol{y}_2 - X_2\boldsymbol{\theta}_{t_2}\|_2^2, \tag{7}$$

with gradient and Hessian

$$\nabla V(\boldsymbol{\theta}) = \begin{bmatrix} X_1^T(X_1\boldsymbol{\theta}_{t_1} - \boldsymbol{y}_1) \\ X_2^T(X_2\boldsymbol{\theta}_{t_2} - \boldsymbol{y}_2) \end{bmatrix}, \qquad H(\boldsymbol{\theta}) = \begin{bmatrix} X_1^T X_1 & 0 \\ 0 & X_2^T X_2 \end{bmatrix}, \tag{8}$$

where

$$X_i = \left[\tilde{\boldsymbol{x}}_j^T\right]_{\boldsymbol{x}_j \in \mathcal{S}_i} \in \mathbb{R}^{|\mathcal{S}_i| \times (d+1)}, \qquad \boldsymbol{y}_i = \left[y_j\right]_{\boldsymbol{x}_j \in \mathcal{S}_i} \in \mathbb{R}^{|\mathcal{S}_i|}, \qquad i = 1, 2,$$

that is, $X_1$ (resp. $X_2$) is the matrix whose rows are the augmented feature vectors $\tilde{\boldsymbol{x}}_j^T$ for all $\boldsymbol{x}_j \in \mathcal{S}_1$ (resp. $\boldsymbol{x}_j \in \mathcal{S}_2$), and $\boldsymbol{y}_1$ (resp. $\boldsymbol{y}_2$) is the vector of the corresponding responses. Here $|\mathcal{S}_i|$ denotes the number of elements in $\mathcal{S}_i$.

The OLS solution for this partition is

$$\boldsymbol{\theta}_{\mathrm{OLS}} = \begin{bmatrix} (X_1^T X_1)^{-1} X_1^T \boldsymbol{y}_1 \\ (X_2^T X_2)^{-1} X_2^T \boldsymbol{y}_2 \end{bmatrix}.$$

As shown in Section 3.2.2,

$$p(\boldsymbol{\theta}) = \boldsymbol{\theta}_{\mathrm{OLS}} - \boldsymbol{\theta} = -H(\boldsymbol{\theta})^{-1} \nabla V(\boldsymbol{\theta}) \tag{9}$$

is the exact Newton/Gauss–Newton direction on this partition. The following assumption is requisite for the node-level convergence.

**Assumption 1** (Regularity). *For every iterate $\boldsymbol{\theta}^{(k)}$:*

(i) (*Nondegenerate hinge*) For all $j$, $\tilde{\boldsymbol{x}}_j^T \boldsymbol{\theta}_{t_1}^{(k)} \neq \tilde{\boldsymbol{x}}_j^T \boldsymbol{\theta}_{t_2}^{(k)}$. Thus no sample lies exactly on the separating hyperplane.

(ii) (*Uniform strong convexity on each fixed partition*) Each block matrix $X_i^T X_i$ encountered along the iterates satisfies

$$mI \preceq X_i^T X_i \preceq LI, \qquad i = 1, 2,$$

for global constants $0 < m \leq L < \infty$.

This condition is standard and matches the leaf-level eigenvalue assumption used in our approximation theory.

The following theorem demonstrates that the line-search Newton update (3) can reach a local optimum of (2).

**Theorem 2** (Node-level convergence of the line-search Newton update). *Under the node-level setup and Assumption 1, consider the Newton iteration at node $\mathbb{D}_t$*

$$\boldsymbol{\theta}^{(k+1)} = \boldsymbol{\theta}^{(k)} + \mu^{(k)} p^{(k)},$$

*where $p^{(k)} = p(\boldsymbol{\theta}^{(k)})$ is the Newton/Gauss–Newton direction given by Equation( 9), and $\mu^{(k)}$ is chosen by a backtracking line search with parameter $\beta \in (0, 1)$ and trial steps $\{\mu_0 \beta^t\}_{t \geq 0}$. Then the following hold:*

(a) *(Per-iterate descent and line-search termination) For any iterate $\boldsymbol{\theta}^{(k)}$ with $\boldsymbol{\theta}^{(k)} \neq \boldsymbol{\theta}_{\mathrm{OLS}}$, the Newton direction $p^{(k)}$ is a strict descent direction and the backtracking line search produces a step size $\mu^{(k)} > 0$ such that*

$$V(\boldsymbol{\theta}^{(k+1)}) = V(\boldsymbol{\theta}^{(k)} + \mu^{(k)} p^{(k)}) < V(\boldsymbol{\theta}^{(k)}). \tag{10}$$

*(b)* *(Monotone decrease and convergence of the objective)* *The sequence $\{V(\boldsymbol{\theta}^{(k)})\}$ is strictly decreasing whenever $p^{(k)} \neq 0$ and converges to a finite limit $V_\infty$.*

*(c)* *(Convergence under a fixed partition)* *If there exists $K$ such that the partition $(\mathcal{S}_1(\boldsymbol{\theta}^{(k)}), \mathcal{S}_2(\boldsymbol{\theta}^{(k)}))$ is constant for all $k \geq K$, then $\{\boldsymbol{\theta}^{(k)}\}$ converges to the unique OLS minimizer for that fixed partition.*

*Proof.* We proceed in several steps.

**Step 1: Newton direction is a strict descent direction.** For any iterate $\boldsymbol{\theta}$ (we temporarily drop the superscript $k$), the Newton direction satisfies

$$p(\boldsymbol{\theta}) = -H(\boldsymbol{\theta})^{-1} \nabla V(\boldsymbol{\theta}),$$

so

$$\nabla V(\boldsymbol{\theta})^T p(\boldsymbol{\theta}) = -\nabla V(\boldsymbol{\theta})^T H(\boldsymbol{\theta})^{-1} \nabla V(\boldsymbol{\theta}).$$

Assumption 1(ii) implies $H(\boldsymbol{\theta}) \preceq LI$ and $H(\boldsymbol{\theta}) \succ 0$, and hence $H(\boldsymbol{\theta})^{-1} \succeq \frac{1}{L}I$. Therefore,

$$\nabla V(\boldsymbol{\theta})^T H(\boldsymbol{\theta})^{-1} \nabla V(\boldsymbol{\theta}) \geq \frac{1}{L} \|\nabla V(\boldsymbol{\theta})\|_2^2,$$

which yields

$$\nabla V(\boldsymbol{\theta})^T p(\boldsymbol{\theta}) \leq -\frac{1}{L} \|\nabla V(\boldsymbol{\theta})\|_2^2.$$

If $\boldsymbol{\theta} \neq \boldsymbol{\theta}_{\mathrm{OLS}}$, then $\nabla V(\boldsymbol{\theta}) \neq 0$, so the right-hand side is strictly negative. Thus $p(\boldsymbol{\theta})$ is a strict descent direction.

**Step 2: Local decrease along $p(\boldsymbol{\theta})$.** Fix $\boldsymbol{\theta}$ and its Newton direction $p = p(\boldsymbol{\theta})$. Write

$$\boldsymbol{\theta} = \begin{bmatrix} \boldsymbol{\theta}_{t_1} \\ \boldsymbol{\theta}_{t_2} \end{bmatrix}, \qquad p(\boldsymbol{\theta}) = \begin{bmatrix} p_1 \\ p_2 \end{bmatrix},$$

conformably with the block structure in Equation( 8).

Consider the univariate function

$$\psi(\mu) := V(\boldsymbol{\theta} + \mu p), \qquad \mu \geq 0.$$

We show that there exists $\tilde{\mu} > 0$ such that $V(\boldsymbol{\theta} + \mu p) < V(\boldsymbol{\theta})$ for all $0 < \mu \leq \tilde{\mu}$.

For each $j$, define

$$a_j(\boldsymbol{\theta}) = \tilde{\boldsymbol{x}}_j^T \boldsymbol{\theta}_{t_1}, \qquad b_j(\boldsymbol{\theta}) = \tilde{\boldsymbol{x}}_j^T \boldsymbol{\theta}_{t_2}.$$

Assumption 1(i) states that $a_j(\boldsymbol{\theta}) \neq b_j(\boldsymbol{\theta})$ for all $j$, i.e., the active branch in the max is unique at $\boldsymbol{\theta}$. Without loss of generality, suppose $a_j(\boldsymbol{\theta}) > b_j(\boldsymbol{\theta})$; the other case is analogous.

Along the ray $\boldsymbol{\theta} + \mu p$ we have

$$a_j(\boldsymbol{\theta} + \mu p) - b_j(\boldsymbol{\theta} + \mu p) = \big(a_j(\boldsymbol{\theta}) - b_j(\boldsymbol{\theta})\big) + \mu\big(\tilde{\boldsymbol{x}}_j^T p_1 - \tilde{\boldsymbol{x}}_j^T p_2\big),$$

which is an affine function of $\mu$ and is nonzero at $\mu = 0$. Therefore, for each $j$ there exists $\bar{\mu}_j > 0$ such that the sign of $a_j(\boldsymbol{\theta} + \mu p) - b_j(\boldsymbol{\theta} + \mu p)$ does not change for all $0 \leq \mu \leq \bar{\mu}_j$. In particular, the active branch in $h(\boldsymbol{x}_j, \cdot)$ remains the same for all small $\mu$.

Let $\bar{\mu} := \min_j \bar{\mu}_j > 0$. Then for all $0 \leq \mu \leq \bar{\mu}$ the index of the active branch in $h(\boldsymbol{x}_j, \cdot)$ is fixed for every $j$, so $V(\boldsymbol{\theta} + \mu p)$ coincides with the smooth quadratic representation (7) along this segment. Hence $\psi(\mu)$ is differentiable on $[0, \bar{\mu}]$ with

$$\psi'(0) = \nabla V(\boldsymbol{\theta})^T p.$$

By Step 1, $\psi'(0) = \nabla V(\boldsymbol{\theta})^T p < 0$. By continuity of $\psi'$ at 0, there exists $0 < \tilde{\mu} \leq \bar{\mu}$ such that

$$\psi(\mu) < \psi(0) = V(\boldsymbol{\theta}) \quad \text{for all } 0 < \mu \leq \tilde{\mu},$$

i.e.,

$$V(\boldsymbol{\theta} + \mu p) < V(\boldsymbol{\theta}) \quad \text{for all } 0 < \mu \leq \tilde{\mu}.$$

**Step 3: Backtracking line search terminates with a decreasing step.** Let the backtracking line search start from some initial step $\mu_0 > 0$ and generate the sequence $\mu_t = \mu_0 \beta^t$ for $t = 0, 1, 2, \dots$ with $\beta \in (0, 1)$. Since $\mu_t \to 0$, there exists $t^\star$ such that $\mu_{t^\star} \leq \tilde{\mu}$. For this $t^\star$ we have $0 < \mu_{t^\star} \leq \tilde{\mu}$ and hence, by Step 2,

$$V(\boldsymbol{\theta} + \mu_{t^\star} p) < V(\boldsymbol{\theta}).$$

The backtracking procedure selects the first such $t^\star$, so it terminates with a step size $\mu^{(k)} = \mu_{t^\star} > 0$ satisfying (10). This proves part (a).

**Step 4: Monotone decrease and convergence of $V$.** Applying part (a) at each iteration $k$ yields

$$V(\boldsymbol{\theta}^{(k+1)}) = V(\boldsymbol{\theta}^{(k)} + \mu^{(k)} p^{(k)}) < V(\boldsymbol{\theta}^{(k)})$$

whenever $p^{(k)} \neq 0$. Since $V(\boldsymbol{\theta}) \geq 0$ for all $\boldsymbol{\theta}$, the sequence $\{V(\boldsymbol{\theta}^{(k)})\}$ is strictly decreasing and bounded below, and thus converges to some finite limit $V_\infty \geq 0$. This establishes part (b).

**Step 5: Convergence under a fixed partition.** Finally, suppose there exists $K$ such that the partition $(\mathcal{S}_1(\boldsymbol{\theta}^{(k)}), \mathcal{S}_2(\boldsymbol{\theta}^{(k)}))$ is constant for all $k \geq K$. Then for all $k \geq K$, the objective $V$ coincides with the smooth, strongly convex quadratic version of (7) with Hessian

$$H(\boldsymbol{\theta}) = \begin{bmatrix} X_1^T X_1 & 0 \\ 0 & X_2^T X_2 \end{bmatrix}, \qquad mI \preceq H(\boldsymbol{\theta}) \preceq LI,$$

by Assumption 1(ii). In this regime, the Newton direction simplifies to

$$p(\boldsymbol{\theta}) = \boldsymbol{\theta}_{\text{OLS}} - \boldsymbol{\theta},$$

and the update becomes

$$\boldsymbol{\theta}^{(k+1)} - \boldsymbol{\theta}_{\text{OLS}} = \left(1 - \mu^{(k)}\right)\left(\boldsymbol{\theta}^{(k)} - \boldsymbol{\theta}_{\text{OLS}}\right).$$

Under the standard backtracking line-search rule on a strongly convex quadratic, the accepted step sizes $\mu^{(k)}$ are uniformly bounded away from zero and from above, so the factor $|1 - \mu^{(k)}|$ is strictly less than 1 and uniformly bounded away from 1. Hence $\boldsymbol{\theta}^{(k)} \to \boldsymbol{\theta}_{\text{OLS}}$ as $k \to \infty$. Equivalently, the iterates converge to the unique OLS minimizer for the fixed partition, which proves part (c) and completes the proof. $\qquad \square$

## C DETAILS ON REGULARIZATION AND FALLBACK STRATEGIES

**Incorporating Ridge Regression (L2 Regularization).** To enhance model robustness and effectively handle situations like multicollinearity or high-dimensional data, we introduce optional ridge regression (L2 regularization) in all OLS fitting steps within this algorithm. ridge regression modifies the OLS objective by adding a penalty on the L2-norm of the model coefficients, i.e., minimizing:

$$\min_{\boldsymbol{\theta}} \frac{1}{2} \sum_{j=1}^{N_k} \left(y_j - \tilde{\boldsymbol{x}}_j^T \boldsymbol{\theta}\right)^2 + \frac{\alpha}{2} \sum_{i=0}^{d-1} \theta_i^2$$

where $\alpha \geq 0$ is the regularization strength parameter. Notably, the bias term $\theta_d$ (corresponding to the constant 1 in the augmented feature vector) is typically not regularized. Given a design matrix $X$ (already augmented with a column of ones for the bias) and a target vector $\mathbf{y}$, the ridge regression solution with regularization parameter $\alpha \geq 0$ is given by:

$$\boldsymbol{\theta} = (X^T X + \alpha I_0)^{-1} X^T \mathbf{y}$$

where $I_0$ is an identity matrix with its last diagonal element (corresponding to the bias term) set to zero, ensuring the bias is not regularized. This regularization mechanism effectively stabilizes parameter estimation, especially when the training data matrix $X^T X$ might be near-singular, which is particularly important in high-dimensional feature spaces or with limited sample sizes. The regularization strength $\alpha$ is tuned as a hyperparameter during model training.

**Fallback Strategy for Non-convergent Nodes.** If the iterative optimization does not converge within $T_{\max}$ iterations, we fall back to a simple median split: a feature dimension $k$ is chosen at random, and the data are split at the median $m_k$ of this dimension. This guarantees progress in tree growth and often decomposes a hard global problem into smaller subproblems that converge quickly under our Newton updates.

# D  PARAMETER INITIALIZATION STRATEGY AND DETAILED ALGORITHMS

This section provides the detailed pseudocode for the optimal node-splitting and tree-building procedures discussed in Section 3.

## D.1  PARAMETER INITIALIZATION STRATEGY

The first step of Algorithms 1 and 2 requires the initialization of the parameter vectors $\theta_1$ and $\theta_2$. A reasonable and stable initialization is important for the convergence of the subsequent alternating optimization. We employ a simple data-driven heuristic combined with a small fallback procedure.

The initialization procedure is as follows:

1. **Heuristic initial partition.** At the current node, we identify the feature dimension with the *largest range* in the local data. We then use the *median* value of this feature as a pivot to split the samples into two subsets. This aims to create an initial partition along the direction of greatest variation, which typically yields a more balanced and meaningful starting point than a fully random split.

2. **Partition-based initial (ridge) regression.** Given these two subsets, we independently solve a (ridge-regularized) least-squares problem on each subset to obtain the initial parameter vectors $\theta_1$ and $\theta_2$. In our implementation we require that each subset contains at least two samples; otherwise we treat the split as too small for a reliable local regression.

3. **Fallback for degenerate or very small partitions.** In edge cases, such as when the node contains very few samples, the features are nearly constant, or one of the subsets after the median split is too small, we fall back to a simple global initialization. Specifically, we first compute a *global* ridge regression solution $\theta_{\text{global}}$ using all samples at the node (or a constant predictor based on the mean response if the design matrix is degenerate). We then construct two initial parameter vectors $\theta_1$ and $\theta_2$ by adding small independent random perturbations to $\theta_{\text{global}}$, which ensures a non-trivial starting point even under poor initial splits.

4. **Ensuring parameter diversity.** Finally, we check whether the resulting $\theta_1$ and $\theta_2$ are nearly identical. If so, we add another small perturbation (and, in a rare corner case, a tiny offset to the first coefficient) to break the symmetry, so that the subsequent alternating optimization can proceed effectively.

This initialization strategy provides a data-dependent yet lightweight starting point for the node-wise optimization, and makes the overall splitting procedure more robust to degenerate partitions and small-sample regimes.

## D.2  DETAILED ALGORITHMS

---

**Algorithm 1** Find Optimal Split (Min Variant)

---

**Require:** Dataset $\mathcal{S} = \{(\boldsymbol{x}_j, y_j)\}_{j=1}^{N}$, Max iterations $T_{\max}$, Step size $\mu$, Tolerance $\epsilon$
**Ensure:** Optimal parameters $\boldsymbol{\theta}_1^*, \boldsymbol{\theta}_2^*$
 1: Initialize $\boldsymbol{\theta}_1^{(0)}, \boldsymbol{\theta}_2^{(0)}$ and partitions $\mathcal{S}_1, \mathcal{S}_2$ using the strategy in Sec. D.1.
 2: **for** $t = 1, \ldots, T_{\max}$ **do**
 3:     *// Model Fitting Step*
 4:     Compute $\boldsymbol{\theta}_{\text{OLS},1}^{(k)}$ and $\boldsymbol{\theta}_{\text{OLS},2}^{(k)}$ **(with optional ridge regularization)**
 5:     $\boldsymbol{\theta}_1^{(k+1)} \leftarrow \boldsymbol{\theta}_1^{(k)} + \mu(\boldsymbol{\theta}_{\text{OLS},1}^{(k)} - \boldsymbol{\theta}_1^{(k)})$                   ▷ Newton update with step size $\mu$
 6:     $\boldsymbol{\theta}_2^{(k+1)} \leftarrow \boldsymbol{\theta}_2^{(k)} + \mu(\boldsymbol{\theta}_{\text{OLS},2}^{(k)} - \boldsymbol{\theta}_2^{(k)})$                   ▷ Newton update with step size $\mu$
 7:     *// Partitioning Step (min: assign to the smaller branch)*
 8:     $\mathcal{S}_{1,\text{new}} \leftarrow \emptyset, \quad \mathcal{S}_{2,\text{new}} \leftarrow \emptyset$
 9:     **for all** $(\boldsymbol{x}_j, y_j) \in \mathcal{S}$ **do**
10:         **if** $\tilde{\boldsymbol{x}}_j^T \boldsymbol{\theta}_1 \leq \tilde{\boldsymbol{x}}_j^T \boldsymbol{\theta}_2$ **then**
11:             $\mathcal{S}_{1,\text{new}} \leftarrow \mathcal{S}_{1,\text{new}} \cup \{(\boldsymbol{x}_j, y_j)\}$
12:         **else**
13:             $\mathcal{S}_{2,\text{new}} \leftarrow \mathcal{S}_{2,\text{new}} \cup \{(\boldsymbol{x}_j, y_j)\}$
14:         **end if**
15:     **end for**
16:     **if** $\|\boldsymbol{\theta}_1^{(k+1)} - \boldsymbol{\theta}_1^{(k)}\| + \|\boldsymbol{\theta}_2^{(k+1)} - \boldsymbol{\theta}_2^{(k)}\| < \epsilon$ **then**     ▷ Check parameter changes
17:         **break**                                             ▷ Convergence
18:     **end if**
19:     $\mathcal{S}_1 \leftarrow \mathcal{S}_{1,\text{new}}, \quad \mathcal{S}_2 \leftarrow \mathcal{S}_{2,\text{new}}$
20: **end for**
21: **return** $\boldsymbol{\theta}_1, \boldsymbol{\theta}_2$

---

**Algorithm 2** Find Optimal Split (Max Variant)

---

**Require:** Dataset $\mathcal{S} = \{(\boldsymbol{x}_j, y_j)\}_{j=1}^{N}$, Max iterations $T_{max}$, Step size $\mu$
**Ensure:** Optimal parameters $\boldsymbol{\theta}_1^*, \boldsymbol{\theta}_2^*$
 1: Initialize $\boldsymbol{\theta}_1^{(0)}, \boldsymbol{\theta}_2^{(0)}$ and partitions $\mathcal{S}_1, \mathcal{S}_2$ using the strategy in Sec. D.1.
 2: **for** $t = 1, \ldots, T_{max}$ **do**
 3:     *// Model Fitting Step*
 4:     Compute $\boldsymbol{\theta}_{\text{OLS},1}^{(k)}$ and $\boldsymbol{\theta}_{\text{OLS},2}^{(k)}$ **(with optional ridge regularization)**
 5:     $\boldsymbol{\theta}_1^{(k+1)} \leftarrow \boldsymbol{\theta}_1^{(k)} + \mu(\boldsymbol{\theta}_{\text{OLS},1}^{(k)} - \boldsymbol{\theta}_1^{(k)})$                   ▷ Newton update with step size $\mu$
 6:     $\boldsymbol{\theta}_2^{(k+1)} \leftarrow \boldsymbol{\theta}_2^{(k)} + \mu(\boldsymbol{\theta}_{\text{OLS},2}^{(k)} - \boldsymbol{\theta}_2^{(k)})$                   ▷ Newton update with step size $\mu$
 7:     *// Partitioning Step (max: assign to the larger branch)*
 8:     $\mathcal{S}_{1,\text{new}} \leftarrow \emptyset, \quad \mathcal{S}_{2,\text{new}} \leftarrow \emptyset$
 9:     **for all** $(\boldsymbol{x}_j, y_j) \in \mathcal{S}$ **do**
10:         **if** $\tilde{\boldsymbol{x}}_j^T \boldsymbol{\theta}_1 \geq \tilde{\boldsymbol{x}}_j^T \boldsymbol{\theta}_2$ **then**
11:             $\mathcal{S}_{1,\text{new}} \leftarrow \mathcal{S}_{1,\text{new}} \cup \{(\boldsymbol{x}_j, y_j)\}$
12:         **else**
13:             $\mathcal{S}_{2,\text{new}} \leftarrow \mathcal{S}_{2,\text{new}} \cup \{(\boldsymbol{x}_j, y_j)\}$
14:         **end if**
15:     **end for**
16:     **if** $\|\boldsymbol{\theta}_1^{(k+1)} - \boldsymbol{\theta}_1^{(k)}\| + \|\boldsymbol{\theta}_2^{(k+1)} - \boldsymbol{\theta}_2^{(k)}\| < \epsilon$ **then**     ▷ Check parameter changes
17:         **break**                                             ▷ Convergence
18:     **end if**
19:     $\mathcal{S}_1 \leftarrow \mathcal{S}_{1,\text{new}}, \quad \mathcal{S}_2 \leftarrow \mathcal{S}_{2,\text{new}}$
20: **end for**
21: **return** $\boldsymbol{\theta}_1, \boldsymbol{\theta}_2$

---

---

**Algorithm 3** Find Optimal Split (Min-or-Max by RMSE)

---

**Require:** Dataset $\mathcal{S} = \{(\boldsymbol{x}_j, y_j)\}_{j=1}^N$, $T_{\max}$, Step size $\mu$, Tolerance $\epsilon$
**Ensure:** Best parameters $(\boldsymbol{\theta}_1^*, \boldsymbol{\theta}_2^*)$, model type $\in \{\min, \max\}$
1: $(\boldsymbol{\theta}_1^{\max}, \boldsymbol{\theta}_2^{\max}) \leftarrow$ run Algorithm 2 (max variant) on $\mathcal{S}$ with $(T_{\max}, \mu, \epsilon)$
2: $(\boldsymbol{\theta}_1^{\min}, \boldsymbol{\theta}_2^{\min}) \leftarrow$ run Algorithm 1 (min variant) on $\mathcal{S}$ with $(T_{\max}, \mu, \epsilon)$
3: Compute $\text{RMSE}_{\max} \leftarrow \sqrt{\frac{1}{N} \sum_{j=1}^N \left(y_j - \max(\tilde{\boldsymbol{x}}_j^T \boldsymbol{\theta}_1^{\max}, \ \tilde{\boldsymbol{x}}_j^T \boldsymbol{\theta}_2^{\max})\right)^2}$
4: Compute $\text{RMSE}_{\min} \leftarrow \sqrt{\frac{1}{N} \sum_{j=1}^N \left(y_j - \min(\tilde{\boldsymbol{x}}_j^T \boldsymbol{\theta}_1^{\min}, \ \tilde{\boldsymbol{x}}_j^T \boldsymbol{\theta}_2^{\min})\right)^2}$
5: **if** $\text{RMSE}_{\min} < \text{RMSE}_{\max}$ **then**
6:     **return** $\boldsymbol{\theta}_1^{\min}, \boldsymbol{\theta}_2^{\min}$
7: **else**
8:     **return** $\boldsymbol{\theta}_1^{\max}, \boldsymbol{\theta}_2^{\max}$
9: **end if**

---

---

**Algorithm 4** Build Oblique Regression Tree

---

**Require:** Dataset $\mathcal{S}$, current depth $d$, hyperparameters (max depth $D_{max}$, min samples $N_{min}$, RMSE threshold $\tau_{RMSE}$, step size $\mu$)

**Ensure:** Root node of a (sub)tree

1: // *Check stopping criteria for creating a leaf*

2: Fit a single linear model for the current node: $\boldsymbol{\theta}_{leaf} \leftarrow \arg\min_{\boldsymbol{\theta}} \sum_{(\boldsymbol{x},y)\in\mathcal{S}} (y - \tilde{\boldsymbol{x}}^T\boldsymbol{\theta})^2$ ▷ Solve via OLS (**with optional ridge regularization**)

3: Calculate RMSE: RMSE $= \sqrt{\frac{1}{|\mathcal{S}|} \sum_{(\boldsymbol{x},y)\in\mathcal{S}} (y - \tilde{\boldsymbol{x}}^T\boldsymbol{\theta}_{leaf})^2}$

4: **if** $d \geq D_{max}$ or $|\mathcal{S}| < N_{min}$ or RMSE $< \tau_{RMSE}$ **then**       ▷ Check depth, samples, or RMSE

5:       Create a leaf node.

6:       Store $\boldsymbol{\theta}_{leaf}$ in the node and **return** the node.

7: **end if**

8:

9: // *Proceed with splitting the node*

10: $\boldsymbol{\theta}_1^*, \boldsymbol{\theta}_2^* \leftarrow$ FindOptimalSplit$(\mathcal{S}, \mu)$

11: $\mathcal{S}_L \leftarrow \{(\boldsymbol{x}, y) \in \mathcal{S} \mid \tilde{\boldsymbol{x}}^T\boldsymbol{\theta}_1^* \geq \tilde{\boldsymbol{x}}^T\boldsymbol{\theta}_2^*\}$

12: $\mathcal{S}_R \leftarrow \mathcal{S} \setminus \mathcal{S}_L$

13:

14: **if** $|\mathcal{S}_L| < N_{min}$ or $|\mathcal{S}_R| < N_{min}$ **then**                        ▷ Ineffective split, create leaf

15:       Create a leaf node.

16:       Store the pre-computed $\boldsymbol{\theta}_{leaf}$ from line 3 and **return** the node.

17: **end if**

18:

19: Create an internal node, store split rule $(\boldsymbol{\theta}_1^*, \boldsymbol{\theta}_2^*)$

20: $N.\text{left\_child} \leftarrow$ BuildTree$(\mathcal{S}_L, d + 1, D_{max}, N_{min}, \tau_{RMSE}, \mu)$

21: $N.\text{right\_child} \leftarrow$ BuildTree$(\mathcal{S}_R, d + 1, D_{max}, N_{min}, \tau_{RMSE}, \mu)$

22: **return** the internal node

---

## E    TIME COMPLEXITY ANALYSIS

### E.1    ALGORITHM 1: FIND OPTIMAL SPLIT

- **Overview:** Algorithm 1 is an iterative algorithm designed to find the optimal splitting parameters $\boldsymbol{\theta}_1$ and $\boldsymbol{\theta}_2$. It alternates between model fitting and partition updating, running for at most $T_{\max}$ iterations.

- **Per Iteration:**
  - **Model Fitting:** Solves OLS for two partitions $\mathcal{S}_1$ and $\mathcal{S}_2$. These OLS solutions can incorporate ridge regression. The complexity of each OLS problem is $O(N_k \cdot d^2)$, where $N_k$ is the number of samples in the partition and $d$ is the feature dimension. Since $N = N_1 + N_2$, the total complexity is $O(N \cdot d^2)$.
  - **Partition Updating:** For each of the $N$ samples, computes two linear functions $\tilde{\boldsymbol{x}}_j^T \boldsymbol{\theta}_1$ and $\tilde{\boldsymbol{x}}_j^T \boldsymbol{\theta}_2$ (each $O(d)$) and reassigns partitions, yielding a complexity of $O(N \cdot d)$.
  - **Total Per Iteration:** $O(N \cdot d^2) + O(N \cdot d) = O(N \cdot d^2)$.

- **Total Iterations:** At most $T_{\max}$, a user-defined maximum.

- **Total Time Complexity:** $O(T_{\max} \cdot N \cdot d^2)$.

- **Note:** In practice, convergence may occur before $T_{\max}$, but the worst-case complexity assumes the full number of iterations.

### E.2    ALGORITHM 2: BUILD OBLIQUE REGRESSION TREE

- **Overview:** Algorithm 2 recursively constructs an oblique regression tree with a maximum depth $D_{\max}$ and a minimum of $N_{\min}$ samples per leaf. Each internal node invokes Algorithm 1, while leaf nodes fit a final linear model.

- **Internal Nodes:**
  - **Splitting:** Each internal node calls Algorithm 1, with complexity $O(T_{\max} \cdot N_{\text{node}} \cdot d^2)$, where $N_{\text{node}}$ is the number of samples at the node.
  - **Worst-Case Assumption:** The tree is balanced, with each split dividing the data approximately in half.
  - **Level Analysis:** At level $k$, there are $2^k$ nodes, each with approximately $N/2^k$ samples. The total complexity per level is:
    $$2^k \cdot O(T_{\max} \cdot (N/2^k) \cdot d^2) = O(T_{\max} \cdot N \cdot d^2).$$
  - **Total Levels:** With $D_{\max}$ levels, the complexity is $O(D_{\max} \cdot T_{\max} \cdot N \cdot d^2)$.

- **Leaf Nodes:**
  - **Fitting:** Each leaf fits an OLS model, which can also incorporate ridge regression, with complexity $O(N_{\text{leaf}} \cdot d^2)$, where $N_{\text{leaf}}$ is the number of samples in the leaf.
  - **Total Leaves:** Up to $O(2^{D_{\max}})$ leaves, but the total sample count across all leaves is $N$. Thus, the total complexity is $O(N \cdot d^2)$.

- **Total Time Complexity:** $O(D_{\max} \cdot T_{\max} \cdot N \cdot d^2) + O(N \cdot d^2) = O(D_{\max} \cdot T_{\max} \cdot N \cdot d^2)$.

- **Note:** In practical settings, $T_{\max}$ and $D_{\max}$ are often fixed constants, simplifying the complexity to $O(N \cdot d^2)$. In theory, if $D_{\max}$ scales with $N$, the complexity increases accordingly.

## F    PROOF OF UNIVERSAL APPROXIMATION AND APPROXIMATION RATE

**Theorem 1.** *Let $\mathcal{F}$ be the class of piecewise linear functions represented by finite oblique regression trees with linear models at the leaves. Let $g : \mathcal{K} \to \mathbb{R}$ be a twice continuously differentiable function ($g \in C^2(\mathcal{K})$) on a compact set $\mathcal{K} \subset \mathbb{R}^d$. Furthermore, assume that for any constructed partition of $\mathcal{K}$ into regions $\{\mathcal{R}_i\}$ with diameter $\leq \delta$, each region $\mathcal{R}_i$ contains $N_i$ training data points $(\boldsymbol{x}_j, g(\boldsymbol{x}_j))_{j=1}^{N_i}$. Let $X_i$ be the $N_i \times (d+1)$ design matrix whose $j$-th row is $\tilde{x}_j^T$. We further assume that $X_i$ satisfies $\lambda_{\min}(X_i^T X_i) \geq c N_i$ for some constant $c > 0$. Also, $\sup_{\boldsymbol{x} \in \mathcal{K}} \|\tilde{\boldsymbol{x}}\|_2 \leq D_{\mathcal{K}}$ for some bounded constant $D_{\mathcal{K}}$. Then, there exists a constant $C$ such that for any $\delta > 0$, we can construct*

*a function $f \in \mathcal{F}$ by partitioning the domain $\mathcal{K}$ into regions $\{\mathcal{R}_i\}$ with $\max_i \operatorname{diam}(\mathcal{R}_i) \leq \delta$, satisfying the uniform error bound:*

$$\sup_{\boldsymbol{x} \in \mathcal{K}} |f(\boldsymbol{x}) - g(\boldsymbol{x})| \leq C\delta^2$$

*This approximation rate directly implies the universal approximation property, i.e., for any $\epsilon > 0$, there exists a function $f \in \mathcal{F}$ such that $\sup_{\boldsymbol{x} \in \mathcal{K}} |f(\boldsymbol{x}) - g(\boldsymbol{x})| < \epsilon$.*

*Proof.* This proof relies on constructing a first-order Taylor approximation within each partition and then quantifying the impact of OLS estimation. The condition that $g$ is twice continuously differentiable ($g \in C^2(\mathcal{K})$) on the compact set $\mathcal{K}$ is crucial throughout this proof.

We adopt the following notation for clarity and consistency: $\boldsymbol{x}$ (or $\boldsymbol{x}_j$) denotes a $d$-dimensional original feature vector. $\tilde{\mathbf{x}}$ (or $\tilde{\mathbf{x}}_j$) denotes the $d+1$-dimensional augmented feature vector $[\boldsymbol{x}^T, 1]^T$ (or $[\boldsymbol{x}_j^T, 1]^T$ respectively), which includes a bias term.

**1. Domain Partitioning.** As is standard, the compact set $\mathcal{K}$ (representing the domain of original feature vectors) can be covered by a finite collection of disjoint convex polytopes $\{\mathcal{R}_i\}_{i=1}^M$ such that $\bigcup_{i=1}^M \mathcal{R}_i = \mathcal{K}$ and the diameter of each polytope $\operatorname{diam}(\mathcal{R}_i) \leq \delta$. Since oblique decision trees perform recursive partitioning with hyperplanes, such a partition can always be realized by a tree structure where the leaf nodes correspond to these regions.

**2. Local Taylor Linear Approximation.** This step introduces a theoretical linear approximation. For each region $\mathcal{R}_i$, we select a center point $\mathbf{c}_i \in \mathcal{R}_i$ and define $f_{\text{Taylor},i}(\boldsymbol{x})$ as the first-order Taylor expansion of $g$ around $\mathbf{c}_i$:

$$f_{\text{Taylor},i}(\boldsymbol{x}) = g(\mathbf{c}_i) + \nabla g(\mathbf{c}_i)^T(\boldsymbol{x} - \mathbf{c}_i) \quad \text{for all } \boldsymbol{x} \in \mathcal{R}_i$$

This $f_{\text{Taylor},i}(\boldsymbol{x})$ is a linear function of $\boldsymbol{x}$. It serves as an analytical benchmark for how well $g(\boldsymbol{x})$ can be approximated locally by a linear function.

**3. Bounding the Taylor Approximation Error (Approximation Error).** By Taylor's theorem with the Lagrange remainder, for any $\boldsymbol{x} \in \mathcal{R}_i$, there exists a point $\boldsymbol{\xi}$ on the line segment between $\boldsymbol{x}$ and $\mathbf{c}_i$ such that:

$$g(\boldsymbol{x}) = g(\mathbf{c}_i) + \nabla g(\mathbf{c}_i)^T(\boldsymbol{x} - \mathbf{c}_i) + \frac{1}{2}(\boldsymbol{x} - \mathbf{c}_i)^T \nabla^2 g(\boldsymbol{\xi})(\boldsymbol{x} - \mathbf{c}_i)$$

where $\nabla^2 g(\boldsymbol{\xi})$ is the Hessian matrix of $g$ at $\boldsymbol{\xi}$. The Taylor approximation error is therefore the magnitude of the remainder term:

$$|f_{\text{Taylor},i}(\boldsymbol{x}) - g(\boldsymbol{x})| = \left| \frac{1}{2}(\boldsymbol{x} - \mathbf{c}_i)^T \nabla^2 g(\boldsymbol{\xi})(\boldsymbol{x} - \mathbf{c}_i) \right|$$

Since $g \in C^2(\mathcal{K})$, its Hessian is bounded on the compact set $\mathcal{K}$. Let $M = \sup_{\mathbf{z} \in \mathcal{K}} \|\nabla^2 g(\mathbf{z})\|_2$ be the supremum of the spectral norm of the Hessian. We can then bound the error:

$$|f_{\text{Taylor},i}(\boldsymbol{x}) - g(\boldsymbol{x})| \leq \frac{1}{2}M\|\boldsymbol{x} - \mathbf{c}_i\|_2^2$$

As both $\boldsymbol{x}$ and $\mathbf{c}_i$ are in $\mathcal{R}_i$, the distance between them is bounded by the diameter, $\|\boldsymbol{x} - \mathbf{c}_i\|_2 \leq \delta$. This yields:

$$|f_{\text{Taylor},i}(\boldsymbol{x}) - g(\boldsymbol{x})| \leq \frac{1}{2}M\delta^2$$

Letting $C_A = M/2$, this term is bounded by $C_A\delta^2$.

**4. Quantification of OLS Estimation Error and its Impact on Convergence Rate.** The preceding analysis assumes we can construct $f_{\text{Taylor},i}(\boldsymbol{x})$ exactly. In practice, however, we only have access to values of $g$ at a finite set of data points. To construct a function $f \in \mathcal{F}$ (as described in the theorem), we use actual data to fit local linear models. Within each region $\mathcal{R}_i$, we employ OLS to estimate a linear model from a set of $N_i$ training data points $\{(\boldsymbol{x}_j, y_j)\}_{j=1}^{N_i}$, where $y_j = g(\boldsymbol{x}_j)$. Let $f_{\text{OLS},i}(\mathbf{x})$ denote this OLS-estimated linear function, which takes the augmented input $\tilde{\mathbf{x}}$.

To quantify the impact of OLS estimation, we decompose the total error for an original input $\boldsymbol{x}$ into two parts:

$$|g(\boldsymbol{x}) - f_{\text{OLS},i}(\mathbf{x})| \leq |g(\boldsymbol{x}) - f_{\text{Taylor},i}(\boldsymbol{x})| + |f_{\text{Taylor},i}(\boldsymbol{x}) - f_{\text{OLS},i}(\mathbf{x})|$$

The first term is the Taylor approximation error, which we have bounded by $C_A\delta^2$. We now need to bound the second term, representing the difference between the OLS-estimated function and the theoretical Taylor function. Letting $f_{\text{Taylor},i}(\boldsymbol{x}) = \mathbf{w}_i^{*T}\tilde{\mathbf{x}}$, where $\mathbf{w}_i^*$ is the corresponding theoretical coefficient vector (i.e., $\mathbf{w}_i^* = [\nabla g(\mathbf{c}_i)^T, g(\mathbf{c}_i) - \nabla g(\mathbf{c}_i)^T \mathbf{c}_i]^T$). The OLS-estimated linear function is $f_{\text{OLS},i}(\mathbf{x}) = \hat{\mathbf{w}}_i^T\tilde{\mathbf{x}}$, where $\hat{\mathbf{w}}_i$ is its coefficient vector.

We know that $g(\boldsymbol{x}_j) = f_{\text{Taylor},i}(\boldsymbol{x}_j) + R(\boldsymbol{x}_j)$, where $|R(\boldsymbol{x}_j)| \leq C_A\delta^2$ is the Taylor remainder. Utilizing the linear model representation, this can be written as $y_j = g(\boldsymbol{x}_j) = \mathbf{w}_i^{*T}\tilde{\mathbf{x}}_j + R(\boldsymbol{x}_j)$ for each data point $(\boldsymbol{x}_j, y_j)$ in $\mathcal{R}_i$.

In matrix-vector notation, for all $N_i$ training data points in $\mathcal{R}_i$, we have:

$$\mathbf{y}_i = X_i\mathbf{w}_i^* + \mathbf{R}_i$$

where $\mathbf{y}_i = [y_1, \ldots, y_{N_i}]^T$ is the vector of observed values, $X_i$ is the $N_i \times (d+1)$ design matrix with rows $\tilde{\mathbf{x}}_j^T$, and $\mathbf{R}_i = [R(\boldsymbol{x}_1), \ldots, R(\boldsymbol{x}_{N_i})]^T$ is the vector of Taylor remainders.

The OLS estimator $\hat{\mathbf{w}}_i$ minimizes the empirical loss $\sum_{j=1}^{N_i}(g(\boldsymbol{x}_j) - \hat{\mathbf{w}}_i^T\tilde{\mathbf{x}}_j)^2$. The solution to this minimization is given by the normal equations:

$$X_i^T X_i\hat{\mathbf{w}}_i = X_i^T\mathbf{y}_i$$

Because $X_i^T X_i$ is invertible (guaranteed by the data assumption in the theorem for sufficiently diverse data), we can solve for $\hat{\mathbf{w}}_i$:

$$\hat{\mathbf{w}}_i = (X_i^T X_i)^{-1}X_i^T\mathbf{y}_i$$

Substituting the expression for $\mathbf{y}_i$ from above into the OLS solution:

$$\hat{\mathbf{w}}_i = (X_i^T X_i)^{-1}X_i^T(X_i\mathbf{w}_i^* + \mathbf{R}_i)$$
$$= (X_i^T X_i)^{-1}X_i^T X_i\mathbf{w}_i^* + (X_i^T X_i)^{-1}X_i^T\mathbf{R}_i$$

Since $(X_i^T X_i)^{-1}X_i^T X_i = I$ (the identity matrix), the equation simplifies to:

$$\hat{\mathbf{w}}_i = \mathbf{w}_i^* + (X_i^T X_i)^{-1}X_i^T\mathbf{R}_i$$

Rearranging this, we obtain the desired expression for the difference:

$$\hat{\mathbf{w}}_i - \mathbf{w}_i^* = (X_i^T X_i)^{-1}X_i^T\mathbf{R}_i$$

Under the data assumption stated in the theorem, we can bound the uniform error of the OLS estimate:

$$\sup_{\boldsymbol{x}\in\mathcal{R}_i} |f_{\text{Taylor},i}(\boldsymbol{x}) - f_{\text{OLS},i}(\mathbf{x})| = \sup_{\boldsymbol{x}\in\mathcal{R}_i} |\tilde{\mathbf{x}}^T(\hat{\mathbf{w}}_i - \mathbf{w}_i^*)|$$
$$\leq \sup_{\boldsymbol{x}\in\mathcal{R}_i} \|\tilde{\mathbf{x}}\|_2 \cdot \|\hat{\mathbf{w}}_i - \mathbf{w}_i^*\|_2$$
$$\leq D_{\mathcal{K}}\|(X_i^T X_i)^{-1}X_i^T\mathbf{R}_i\|_2$$
$$\leq D_{\mathcal{K}}\|(X_i^T X_i)^{-1}X_i^T\|_2\|\mathbf{R}_i\|_2$$

From the theorem's assumption, $\lambda_{\min}(X_i^T X_i) \geq cN_i$ for some constant $c > 0$. This implies that the smallest singular value of $X_i$, $\sigma_{\min}(X_i) = \sqrt{\lambda_{\min}(X_i^T X_i)} \geq \sqrt{cN_i}$. The spectral norm of the pseudoinverse $X_i^+ = (X_i^T X_i)^{-1}X_i^T$ is given by $\|X_i^+\|_2 = \frac{1}{\sigma_{\min}(X_i)}$. Therefore, we have:

$$\|(X_i^T X_i)^{-1}X_i^T\|_2 \leq \frac{1}{\sqrt{cN_i}}$$

Substituting this into the bound:

$$\sup_{\boldsymbol{x}\in\mathcal{R}_i} |f_{\text{Taylor},i}(\boldsymbol{x}) - f_{\text{OLS},i}(\mathbf{x})| \leq D_{\mathcal{K}} \cdot \frac{1}{\sqrt{cN_i}} \cdot \|\mathbf{R}_i\|_2$$

Since $\|\mathbf{R}_i\|_2 \leq \sqrt{N_i} \sup_j |R(\boldsymbol{x}_j)| \leq \sqrt{N_i} C_A \delta^2$, we obtain:

$$\sup_{\boldsymbol{x} \in \mathcal{R}_i} |f_{\text{Taylor},i}(\boldsymbol{x}) - f_{\text{OLS},i}(\mathbf{x})| \leq D_\mathcal{K} \frac{1}{\sqrt{cN_i}} \sqrt{N_i} C_A \delta^2 = \frac{D_\mathcal{K} C_A}{\sqrt{c}} \delta^2$$

This term is of order $O(\delta^2)$, with a constant factor $C_O = D_\mathcal{K} C_A / \sqrt{c}$. This indicates that as long as the data density and diversity are sufficient (ensuring $N_i$ is large enough and $c$ is well above zero), OLS can converge to the theoretically optimal local linear approximation, and the additional error it introduces does not degrade the $O(\delta^2)$ convergence rate.

**Combining Total Error:** Combining both error components, for each region $\mathcal{R}_i$ and any original feature vector $\boldsymbol{x} \in \mathcal{R}_i$, we have:

$$|g(\boldsymbol{x}) - f_{\text{OLS},i}(\mathbf{x})| \leq C_A \delta^2 + C_O \delta^2 = (C_A + C_O) \delta^2$$

Let $C = C_A + C_O = C_A(1 + D_\mathcal{K}/\sqrt{c})$. We define the constructed piecewise linear function $f \in \mathcal{F}$ such that for $\boldsymbol{x} \in \mathcal{R}_i$, $f(\boldsymbol{x}) = f_{\text{OLS},i}(\mathbf{x})$. Thus, this bound holds for the entire domain $\mathcal{K}$:

$$\sup_{\boldsymbol{x} \in \mathcal{K}} |g(\boldsymbol{x}) - f(\boldsymbol{x})| \leq C \delta^2$$

This demonstrates that even with OLS for local linear fitting, provided sufficient data $N_i$ in each region to stabilize the OLS estimates, the $O(\delta^2)$ convergence rate is maintained. The universal approximation property also follows. $\square$

**Corollary 1** (Curse of Dimensionality). To achieve a partition with a maximum region diameter of $\delta$, the depth of a well-balanced tree, $D_{\max}$, scales with the dimension $d$. To halve a region's diameter, one must ideally cut it along all $d$ orthogonal directions, which requires approximately $d$ splits, or $d$ levels in the tree. This leads to the approximate relationship:

$$\delta(D_{\max}) \approx \text{diam}(\mathcal{K}) \cdot 2^{-D_{\max}/d}$$

This implies that achieving a fixed error level (i.e., a fixed $\delta$) requires the tree depth $D_{\max}$ to grow linearly with dimension $d$ ($D_{\max} \propto d \log(1/\delta)$), a manifestation of the curse of dimensionality inherent to space-partitioning methods.

# G  ABLATION STUDY ON STEP SIZE: STABILITY, EFFICIENCY, AND FALLBACK

The step size $\mu$ (damping factor) is a key hyperparameter in our node-splitting algorithm. It controls how aggressively each Newton update moves toward the local OLS solution and therefore directly affects stability, efficiency, and final performance. To examine these effects empirically, we perform an ablation over $\mu$ on four datasets with different characteristics: the oscillatory `sinc` function, the smoother `twisted_sigmoid` function, and the real-world `Kinematics` and `D-Ailerons` datasets. For the two real-world datasets, we additionally evaluate an automatic line-search step-size strategy (denoted by `auto`). Results are reported in Tables 5–8.

**1. Damping and the fallback mechanism.** On the synthetic datasets, smaller step sizes lead to very few fallbacks, while more aggressive steps can trigger the fallback mechanism more often. For `sinc`, the fallback rate is essentially negligible at $\mu = 0.01$ (**0.39%**), modest at $\mu = 0.05$ (**4.74%**), and rises to **32.86%**, **60.94%**, and **29.55%** at $\mu = 0.10, 0.50$, and $1.00$, respectively. A similar but milder trend appears on `twisted_sigmoid`: the fallback rate increases from **0.67%** at $\mu = 0.01$ to **34.67%** at $\mu = 0.50$, before dropping to **19.33%** at $\mu = 1.00$. On the real-world datasets, fallback rates are moderate and relatively stable across $\mu$ for `Kinematics` (between **6.87%** and **10.82%**, with **9.51%** for `auto`), and identically zero on `D-Ailerons` for all settings, including `auto`. Overall, these results indicate that damping does help keep the fallback mechanism rare on more challenging synthetic objectives, while some benign regression problems remain stable even for relatively aggressive step sizes.

**2. Step size, iterations, and training time.**  Larger step sizes do not automatically translate into faster convergence in wall-clock time. On the `Kinematics` datasets, small or moderately damped step sizes typically lead to stable and fairly regular Newton dynamics, so the node-wise optimisation terminates after only a few iterations on most splits. In contrast, aggressive step sizes can induce oscillatory or irregular behaviour at some nodes: the objective value may fail to decrease monotonically, and the parameters and partitions can keep moving without settling down quickly. Such oscillatory runs require many more iterations before the stopping criterion is met, which substantially increases both the average iteration count and the total training time. As a result, the dependence of fit time on $\mu$ is non-monotone: large steps can be very fast when they converge cleanly, but they may be markedly less efficient once the cost of these unstable cases is taken into account.

**3. Effect on predictive accuracy.**  The impact of $\mu$ on RMSE is dataset dependent but generally modest, with a few notable cases. On `sinc`, the smallest step size $\mu = 0.01$ achieves the lowest error (**0.02796**), while $\mu = 0.10$ is clearly suboptimal (**0.05310**); larger steps ($\mu = 0.50$ and $1.00$) recover competitive errors but at the cost of higher fallback rates and iteration counts. On `twisted_sigmoid`, all choices of $\mu$ yield similar performance, with a slight advantage around $\mu = 0.50$ and $1.00$ (RMSE $\approx$ **0.0258–0.0283**). On `Kinematics`, performance improves as $\mu$ increases up to $0.50$ (RMSE decreasing from **0.13055** at $\mu = 0.01$ to **0.10266** at $\mu = 0.50$), and the `auto` strategy yields the lowest error (**0.10185**) while using far fewer iterations than $\mu = 0.50$. On `D-Ailerons`, all step-size choices, including `auto`, achieve essentially identical RMSE (about $1.68 \times 10^{-4}$), indicating that accuracy is largely insensitive to the damping factor on this dataset.

**4. Overall takeaway.**  Taken together, these experiments suggest that the damping factor $\mu$ mainly controls a trade-off between stability, efficiency, and the need for fallbacks, with only moderate influence on accuracy in most settings. Small or moderate fixed step sizes tend to keep fallbacks rare and iterations moderate, especially on synthetic problems such as `sinc` and `twisted_sigmoid`. Extremely aggressive fixed steps can lead to increased fallback usage and substantially more iterations, particularly on more structured or high-dimensional data such as `Kinematics`. The automatic line-search step size (`auto`), which corresponds to the theoretically analysed variant in Appendix B, consistently provides a robust choice on the real-world datasets: it attains strong accuracy, moderate iteration counts, and stable fallback behaviour without requiring manual tuning of $\mu$.

It is important to note that Figures 1 and 2 focus on the dynamics of a single internal node under different step sizes, with a fixed initialization. In contrast, Tables 5–8 report averages over all nodes in a full tree, including the effect of the fallback mechanism and stochastic variation across runs. As a result, local node-level stability and global tree-level efficiency are related but not identical quantities, and their trends need not coincide pointwise for every value of $\mu$.

Table 5: Ablation study on the `sinc` dataset. Metrics are averaged over 10 runs.

| Step size ($\mu$) | RMSE ↓ | Leaves | Avg. Iters | Fit Time (s) | Avg. Fallbacks | Avg. Splits | Fallback Rate (%) |
|---|---|---|---|---|---|---|---|
| 0.01 | 0.02796 | 26.7 | 2.56 | 0.166 | 0.1 | 25.7 | 0.39% |
| 0.05 | 0.02885 | 24.2 | 4.46 | 0.105 | 1.1 | 23.2 | 4.74% |
| 0.10 | 0.05310 | 22.3 | 5.97 | 0.107 | 7.0 | 21.3 | 32.86% |
| 0.50 | 0.03069 | 26.6 | 4.30 | 0.112 | 15.6 | 25.6 | 60.94% |
| 1.00 | 0.02926 | 23.0 | 6.65 | 0.141 | 6.5 | 22.0 | 29.55% |

Table 6: Ablation study on the `twisted_sigmoid` dataset. Metrics are averaged over 10 runs.

| Step size ($\mu$) | RMSE ↓ | Leaves | Avg. Iters | Fit Time (s) | Avg. Fallbacks | Avg. Splits | Fallback Rate (%) |
|---|---|---|---|---|---|---|---|
| 0.01 | 0.02653 | 16.0 | 3.17 | 0.045 | 0.1 | 15.0 | 0.67% |
| 0.05 | 0.03011 | 16.0 | 4.44 | 0.083 | 0.6 | 15.0 | 4.00% |
| 0.10 | 0.02831 | 15.6 | 5.28 | 0.063 | 3.0 | 14.6 | 20.55% |
| 0.50 | 0.02578 | 16.0 | 6.13 | 0.107 | 5.2 | 15.0 | 34.67% |
| 1.00 | 0.02577 | 16.0 | 7.54 | 0.121 | 2.9 | 15.0 | 19.33% |

Table 7: Ablation study on the `Kinematics` dataset. Metrics are averaged over five runs.

| Step size ($\mu$) | RMSE ↓ | Leaves | Avg. Iters | Fit Time (s) | Avg. Fallbacks | Avg. Splits | Fallback Rate (%) |
|---|---|---|---|---|---|---|---|
| 0.01 | 0.13055 | 57.0 | 10.89 | 0.49 | 4.6 | 56.0 | 8.21% |
| 0.05 | 0.11367 | 59.2 | 19.53 | 0.62 | 4.0 | 58.2 | 6.87% |
| 0.10 | 0.10631 | 57.2 | 26.65 | 0.72 | 6.0 | 56.2 | 10.68% |
| 0.50 | 0.10266 | 56.6 | 59.77 | 1.53 | 5.8 | 57.2 | 10.14% |
| 1.00 | 0.10702 | 48.6 | 83.43 | 1.88 | 5.8 | 53.6 | 10.82% |
| auto | 0.10185 | 48.6 | 15.76 | 0.79 | 5.4 | 56.8 | 9.51% |

Table 8: Ablation study on the `D-Ailerons` dataset. Metrics are averaged over five runs. The reported RMSE values are scaled by $\times 10^{-4}$.

| Step size ($\mu$) | RMSE ↓ | Leaves | Avg. Iters | Fit Time (s) | Avg. Fallbacks | Avg. Splits | Fallback Rate (%) |
|---|---|---|---|---|---|---|---|
| 0.01 | 1.68 | 8 | 1.57 | 0.105 | 0 | 7 | 0% |
| 0.05 | 1.68 | 8 | 32.43 | 0.178 | 0 | 7 | 0% |
| 0.10 | 1.68 | 8 | 30.09 | 0.179 | 0 | 7 | 0% |
| 0.50 | 1.68 | 8 | 57.69 | 0.345 | 0 | 7 | 0% |
| 1.00 | 1.68 | 8 | 92.60 | 0.525 | 0 | 7 | 0% |
| auto | 1.68 | 8 | 6.71 | 0.183 | 0 | 7 | 0% |

## H   SYNTHETIC DATA DESCRIPTION

This appendix provides the detailed mathematical definitions of the 3D synthetic functions used in Section 5.2, the rationale for the selection of these synthetic functions and noise levels, and additional visualizations of our method's 2D and 3D function approximation capabilities (Figures 3 and 4).

For all 3D functions, the inputs $x_1, x_2$ range from $[-3, 3]$.

**3D Oscillatory Surface Functions:**

- $f_1(x_1, x_2) = \left(0.5\, x_1^3 - 2\, x_1 x_2^2 + 3 \sin\left(4x_1\right) \cos\left(2x_2\right) + 0.1\, e^{-\left(x_1^2 + x_2^2\right)}\right)$

- $f_2(x_1, x_2) = \sin(3x_1) + \cos(2x_2) + 0.5\, \sin(5x_1)\cos(4x_2)$

- $f_3(x_1, x_2) = \frac{x_1^2 - x_2^2}{0.5 + r^2} + \sin(r)e^{-r}$, where $r = \sqrt{x_1^2 + x_2^2} + 10^{-6}$.

- $f_4(x_1, x_2) = 2 \cdot \exp\left(-\frac{(x_1-1)^2 + (x_2-1)^2}{0.5}\right) - 3 \cdot \exp\left(-\frac{(x_1+1)^2 + (x_2+1.5)^2}{0.3}\right) + 0.5\, x_1$

**Rationale for Selection of Synthetic Functions:** These functions exhibit diverse nonlinear characteristics, including high-frequency oscillations (`sinc`, 3D oscillatory functions), sharp changes, and inflection points (`twisted_sigmoid`). Successfully approximating such a wide range of nonlinearities, as demonstrated by our empirical results, provides strong empirical support for the universal approximation capability of our piecewise linear HRT model, which is theoretically proven in Theorem 1. The `sinc` function, with its numerous local extrema and oscillations, presents a particularly challenging landscape for optimization algorithms. It helps to expose potential instabilities (e.g., non-monotonic convergence, limit cycles) in the splitting mechanism, thereby highlighting the necessity and effectiveness of damped Newton updates for robust convergence, as analyzed in Section 5.1. Conversely, the `twisted_sigmoid` function, while non-trivial, is smoother and less prone to pathological behaviors, allowing us to demonstrate the efficiency of our Newton updates in a more stable environment.

**Rationale for Noise Levels:** Introducing independent and identically distributed zero-mean Gaussian noise (with standard deviations of $0.025$ for 2D tasks and $0.05$ for 3D tasks) simulates this inherent uncertainty. This ensures that our model's performance is evaluated not just on ideal, noiseless functions, but on data that more closely mimic practical applications. The selected noise levels are not so high that the primary challenge for the models remains the accurate approximation of the underlying functional relationship, rather than solely distinguishing signal from overwhelming noise.

This allows us to clearly observe and compare the models' capacities to fit the true, underlying non-linear structures. If noise were excessively high, it would obscure the approximation performance, making it difficult to assess the inherent expressive power of the models.

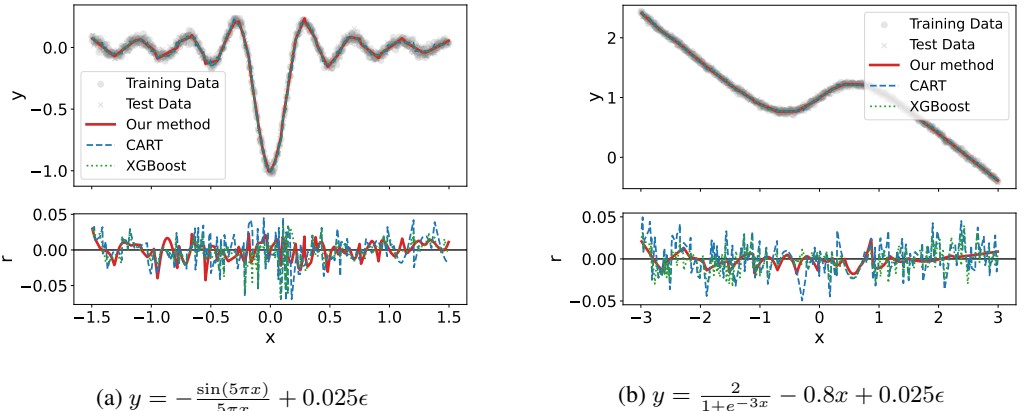

(a) $y = -\frac{\sin(5\pi x)}{5\pi x} + 0.025\epsilon$

(b) $y = \frac{2}{1+e^{-3x}} - 0.8x + 0.025\epsilon$

Figure 3: Top: Approximation performance of various methods on `sinc` and `twisted_sigmoid` functions. Bottom: Residuals $r = y_{\text{pred}} - y_{\text{true}}$, representing the difference between predicted and true values for each method.

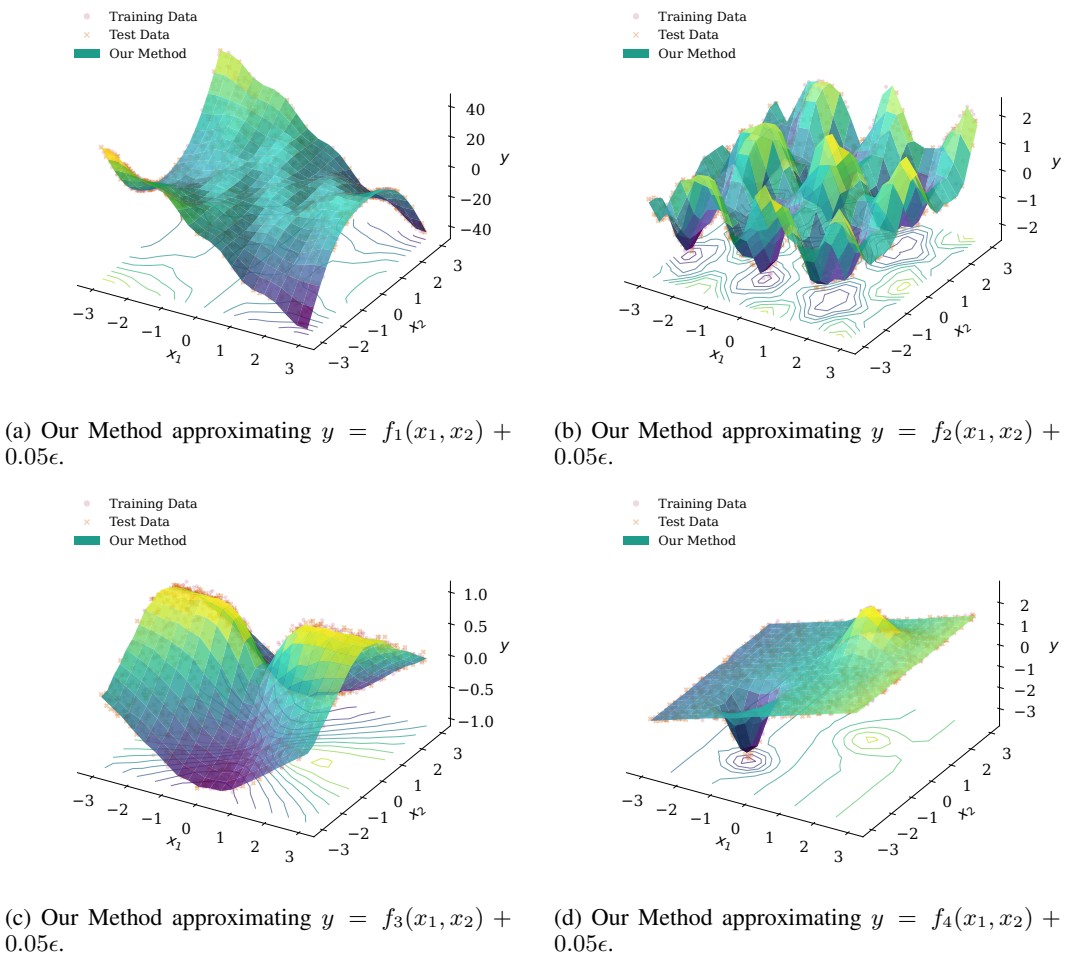

(a) Our Method approximating $y = f_1(x_1, x_2) + 0.05\epsilon$.

(b) Our Method approximating $y = f_2(x_1, x_2) + 0.05\epsilon$.

(c) Our Method approximating $y = f_3(x_1, x_2) + 0.05\epsilon$.

(d) Our Method approximating $y = f_4(x_1, x_2) + 0.05\epsilon$.

Figure 4: 3D function approximation. We visualize the learned piecewise linear surface of our method along with training/testing points. Contours on the floor help read depth. For clarity, only the fitted surface of Our Method is shown; detailed quantitative comparisons with baselines are provided in Table 1.

.

## I   HYPERPARAMETER

This appendix details the hyperparameter search grids used for tuning various models in 2D and 3D function approximation experiments. For each model, a grid search approach was employed to find the optimal hyperparameters that minimize the validation error. The specific ranges for each hyperparameter are listed in Table 9. Hyperparameters are tuned via five-fold cross-validation on the training set, with final performance reported on the testing set.

Table 9: Hyperparameter Search Grids for 3D Function Approximation Models

| Model | Hyperparameter | Search Grid |
|---|---|---|
| HRT | `max_depth` | $\{4, 6, 8, 10, 12\}$ |
| | `threshold` | $\{0.01, 0.03, 0.05\}$ |
| | `step_size` | $\{0.01, 0.5, 1, \text{'auto'}\}$ |
| | `ridge_alpha` | $\{0.001, 0, 1\}$ |
| CART | `max_depth` | $\{3, 5, 7, 9, 11\}$ |
| | `min_samples_split` | $\{2, 5, 10\}$ |
| | `min_samples_leaf` | $\{1, 2, 4\}$ |
| XGB | `n_estimators` | $\{50, 100, 200\}$ |
| | `learning_rate` | $\{0.01, 0.1, 0.2\}$ |
| | `max_depth` | $\{2, 3, 5\}$ |
| | `subsample` | $\{0.7, 1.0\}$ |

For HRT, we tuned the `threshold` parameter, which controls the splitting criterion, and `max_depth`, which limits the maximum depth of the constructed tree structure. We also varied the `ridge_alpha`, which adjusts the regularization strength in ridge regression to mitigate overfitting, and the `step_size`, which determines the granularity of updates during optimization and affects convergence speed and stability. For CART, we explored variations in `max_depth`, the minimum number of samples required to split an internal node (`min_samples_split`), and the minimum number of samples required to be at a leaf node (`min_samples_leaf`). For XGBoost, we optimized the number of boosting rounds (`n_estimators`), the step size shrinkage (`learning_rate`), the maximum depth of a tree (`max_depth`), and the subsample ratio of the training instance (`subsample`). The objective function used for XGBoost was `'reg:squarederror'` with `'rmse'` as the evaluation metric.

Tables 10 and 11 summarize the chosen hyperparameters via 5-fold cross-validation, covering both our proposed method and all reproduced baseline methods. The tuned hyperparameters for DTSem-Net and DGT are listed in Table 12,.

Table 10: Optimal Hyperparameters Across Various Functions for Different Models

| Function | Model | Best Hyperparameters |
|---|---|---|
| sinc | CART | {'max_depth': 9, 'min_samples_leaf': 2, 'min_samples_split': 2} |
| | XGB | {'learning_rate': 0.1, 'max_depth': 3, 'n_estimators': 200, 'subsample': 0.7} |
| | HRT | {'max_depth': 6, 'ridge_alpha': 0.001, 'step_size': 0.01, 'threshold': 0.03} |
| twisted_sigmoid | CART | {'max_depth': 9, 'min_samples_leaf': 2, 'min_samples_split': 5} |
| | XGB | {'learning_rate': 0.1, 'max_depth': 3, 'n_estimators': 100, 'subsample': 0.7} |
| | HRT | {'max_depth': 4, 'ridge_alpha': 0.001, 'step_size': 0.5, 'threshold': 0.01} |
| $f_1(x_1, x_2)$ | CART | {'max_depth': 11, 'min_samples_leaf': 1, 'min_samples_split': 2} |
| | XGB | {'learning_rate': 0.2, 'max_depth': 5, 'n_estimators': 200, 'subsample': 0.7} |
| | HRT | {'max_depth': 12, 'threshold': 0.01, 'ridge_alpha': 0, 'step_size': 1} |
| $f_2(x_1, x_2)$ | CART | {'max_depth': 11, 'min_samples_leaf': 1, 'min_samples_split': 2} |
| | XGB | {'learning_rate': 0.2, 'max_depth': 5, 'n_estimators': 200, 'subsample': 0.7} |
| | HRT | {'max_depth': 12, 'threshold': 0.01, 'ridge_alpha': 0, 'step_size': 1} |
| $f_3(x_1, x_2)$ | CART | {'max_depth': 11, 'min_samples_leaf': 4, 'min_samples_split': 2} |
| | XGB | {'learning_rate': 0.1, 'max_depth': 5, 'n_estimators': 200, 'subsample': 0.7} |
| | HRT | {'max_depth': 8, 'threshold': 0.05, 'ridge_alpha': 0, 'step_size': 1} |
| $f_4(x_1, x_2)$ | CART | {'max_depth': 11, 'min_samples_leaf': 1, 'min_samples_split': 2} |
| | XGB | {'learning_rate': 0.1, 'max_depth': 5, 'n_estimators': 200, 'subsample': 0.7} |
| | HRT | {'max_depth': 12, 'threshold': 0.05, 'ridge_alpha': 0, 'step_size': 1} |

Table 11: Optimal Hyperparameters Across Various Datasets for Different Models

| Dataset | Model | Best Hyperparameters |
|---|---|---|
| Abalone | M5 | `{'M': 10.0}` |
| | Linear tree | `{'max_depth': 3, 'min_samples_leaf': 40}` |
| | HRT | `{'max_depth': 2, 'ridge_alpha': 0, 'step_size': 1, 'threshold': 1}` |
| CPUact | M5 | `{'M': 4.0}` |
| | Linear tree | `{'max_depth': 3, 'min_samples_leaf': 10}` |
| | HRT | `{'max_depth': 2, 'ridge_alpha': 0, 'step_size': 0.5, 'threshold': 0.5}` |
| Ailerons | M5 | `{'M': 40.0 }` |
| | Linear tree | `{'max_depth': 3, 'min_samples_leaf': 10}` |
| | HRT | `{'max_depth': 2, 'ridge_alpha': 1, 'step_size': 1, 'threshold': 5e-05}` |
| CTSlice | M5 | `{'M': 20.0}` |
| | Linear tree | `{'max_depth': 5, 'min_samples_leaf': 10}` |
| | HRT | `{'max_depth': 7, 'ridge_alpha': 10, 'step_size': 1, 'threshold': 0.2}` |
| YearPred | M5 | `{'M': 200.0}` |
| | Linear tree | `{ max_depth=5, min_samples_leaf=2000 }` |
| | HRT | `{'max_depth': 4, 'ridge_alpha': 0, 'step_size': 'auto', 'threshold': 0.5}` |
| Concrete | XGB | `{'learning_rate': 0.01, 'max_depth': 7, 'n_estimators': 200, 'subsample': 0.7}` |
| | M5 | `{'M': 4.0}` |
| | Linear tree | `{'max_depth': 3, 'min_samples_leaf': 40}` |
| | HRT | `{'max_depth': 3, 'ridge_alpha': 0.1, 'step_size': 0.5, 'threshold': 6}` |
| Airfoil | XGB | `{'learning_rate': 0.01, 'max_depth': 7, 'n_estimators': 200, 'subsample': 0.7}` |
| | M5 | `{ M': 4.0}` |
| | Linear tree | `{max_depth': 6, 'min_samples_leaf': 50 }` |
| | HRT | `{'max_depth': 5, 'ridge_alpha': 0.01, 'step_size': 'auto', 'threshold': 1.5}` |
| Fried | CART | `{'max_depth': 11, 'min_samples_leaf': 4, 'min_samples_split': 10}` |
| | XGB | `{'learning_rate': 0.1, 'max_depth': 5, 'n_estimators': 200, 'subsample': 0.7}` |
| | M5 | `{'M': 4.0}` |
| | Linear tree | `{'max_depth': 5, 'min_samples_leaf': 10}` |
| | HRT | `{'max_depth': 5, 'ridge_alpha': 0.1, 'step_size': 0.1, 'threshold': 0}` |
| D-Elevators | CART | `{'max_depth': 5, 'min_samples_leaf': 2, 'min_samples_split': 10}` |
| | XGB | `{'learning_rate': 0.01, 'max_depth': 5, 'n_estimators': 200, 'subsample': 0.7}` |
| | M5 | `{'M': 20.0}` |
| | Linear tree | `{'max_depth': 5, 'min_samples_leaf': 20}` |
| | HRT | `{'max_depth': 5, 'ridge_alpha': 10, 'step_size': 1, 'threshold': 0}` |
| D-Ailerons | CART | `{'max_depth': 5, 'min_samples_leaf': 4, 'min_samples_split': 10}` |
| | XGB | `{'learning_rate': 0.1, 'max_depth': 3, 'n_estimators': 200, 'subsample': 0.7}` |
| | M5 | `{'M': 20.0}` |
| | Linear tree | `{'max_depth': 3, 'min_samples_leaf': 20}` |
| | HRT | `{'max_depth': 3, 'ridge_alpha': 10, 'step_size': 'auto', 'threshold': 0}` |
| Kinematics | CART | `{'max_depth': 9, 'min_samples_leaf': 4, 'min_samples_split': 10}` |
| | XGB | `{'learning_rate': 0.1, 'max_depth': 7, 'n_estimators': 200, 'subsample': 0.7}` |
| | M5 | `{'M': 4.0}` |
| | Linear tree | `{'max_depth': 7, 'min_samples_leaf': 40}` |
| | HRT | `{'max_depth': 6, 'ridge_alpha': 1, 'step_size': 'auto', 'threshold': 0}` |
| C&C | CART | `{'max_depth': 3, 'min_samples_leaf': 4, 'min_samples_split': 2}` |
| | XGB | `{'learning_rate': 0.01, 'max_depth': 5, 'n_estimators': 200, 'subsample': 0.7}` |
| | M5 | `{'M': 40.0}` |
| | Linear tree | `{'max_depth': 9, 'min_samples_leaf': 10}` |
| | HRT | `{'max_depth': 1, 'ridge_alpha': 300, 'step_size': 0.01, 'threshold': 0.0}` |
| Blog | CART | `{'max_depth': 5, 'min_samples_leaf': 1, 'min_samples_split': 2}` |
| | XGB | `{'learning_rate': 0.05, 'max_depth': 3, 'n_estimators': 400}` |
| | M5 | `{'M': 20.0}` |
| | HRT | `{'max_depth': 2, 'ridge_alpha': 100, 'step_size': 0.01, 'threshold': 0}` |

Table 12: Hyperparameters for DGT and DTSemNet

| Parameters | DTSemNet | DGT |
|---|---|---|
| Height | 5 | 6 |
| Learning Rate | 0.005 | 0.01 |
| Batch Size | 32 | 128 |
| Num Epochs | 80 | 200 |
| Optimizer | Adam | RMSprop |

## J   DATASET DETAILS

• **Abalone:** This dataset is designed to predict the age of abalone, given eight features including categorical and numerical measurements describing physical characteristics.

• **CPUact:** This dataset is designed to predict the portion of time that CPUs run in user mode, given numerical features describing system performance measurements.

• **Ailerons:** This dataset is designed to predict the aileron control command, given numerical features describing the status of the aircraft.

• **CTSlice:** This dataset is designed to predict the relative location of CT slices on the axial axis of the human body, given numerical features describing bone and air distribution patterns extracted from CT images.

• **YearPred:** This dataset is designed to predict the release year of a song, given numerical features describing audio characteristics.

• **Concrete:** This dataset is designed to predict the compressive strength of concrete, given eight quantitative mixture components together with the age of the concrete as input features.

• **Airfoil:** This dataset is designed to predict the scaled sound pressure level of airfoils, given five numerical features describing aerodynamic and geometric properties.

• **Fried:** This dataset is designed to predict a continuous target variable, given ten numerical features generated independently and uniformly.

• **D-Elevators:** This dataset is designed to predict the variation of elevator control signals for an F16 aircraft, given six numerical features describing the aircraft state.

• **D-Ailerons:** This dataset is designed to predict the variation of aileron control signals for an F16 aircraft, given numerical features describing the aircraft state.

• **Kinematics:** This dataset is designed to predict the forward kinematics of an 8-link robot arm, given numerical features describing joint configurations. The variant used (8nm) is highly nonlinear and moderately noisy.

• **C&C (Communities & Crime):** This dataset combines census, law-enforcement, and crime statistics to predict community crime rates. It is high-dimensional and heterogeneous, serving as a standard benchmark for socio-demographic prediction.

• **Blog:** This dataset predicts the number of comments on blog posts using text-derived features. It features a skewed target distribution and nonlinear relationships, making it suitable for modeling online content influence.

Table 13: Dataset Details

| Dataset | features | Total sample number | Source |
|---|---|---|---|
| Abalone | 8 | 4177 | UCI[1] |
| CPUact | 21 | 8192 | Delve[2] |
| Ailerons | 40 | 13750 | LIACC[3] |
| CTSlice | 384 | 53500 | UCI[1] |
| YearPred | 90 | 515345 | UCI[1] |
| Concrete | 8 | 1030 | UCI[1] |
| Airfoil | 5 | 1503 | UCI[1] |
| Fried | 10 | 40768 | LIACC[3] |
| D-Elevators | 6 | 9517 | LIACC[3] |
| D-Ailerons | 5 | 7129 | LIACC[3] |
| Kinematics | 8 | 8192 | LIACC[3] |
| C&C | 127 | 1994 | UCI[1] |
| Blog | 280 | 60021 | UCI[1] |

## K    EXTENDING HRT TO BINARY CLASSIFICATION

Although HRT was originally developed for regression tasks, it naturally extends to binary classification. We treat the binary labels $\{0, 1\}$ directly as continuous regression targets, allowing the model to learn locally optimal linear decision functions at each leaf. At inference, we interpret the tree output $\hat{y}(x)$ as a real-valued score, clip it to $[0, 1]$ to obtain a class-probability estimate, and predict the positive class if $\hat{y}(x) \geq 0.5$ and the negative class otherwise.

To assess the classification capability of HRT, we evaluated it on five public binary classification datasets (Breast-w, Credit-g, Banknote, Spambase, and Diabetes), comparing its performance with that of CART and XGBoost. Across these datasets, HRT achieves AUC, Accuracy, and F1 scores that are broadly competitive with—and in several cases exceed—those of XGBoost, while requiring substantially fewer leaf nodes. This reduction in model complexity contributes to improved interpretability without sacrificing predictive accuracy. The detailed results (mean ± standard deviation over 5 random repetitions) are reported in Table 14.

To ensure a fair and consistent comparison, we conducted independent hyperparameter tuning for each model on every dataset. Although XGBoost constitutes a strong ensemble baseline due to its use of multiple boosted trees, HRT employs only a single-tree structure. Despite this difference in model complexity, the empirical results indicate that HRT can reach competitive levels of predictive performance while maintaining a simpler and more transparent model form. The best hyperparameter configurations selected for all models are summarized in Table 15.

Table 14: Performance comparison on five real-world datasets. Values for AUC, ACC, and F1 are reported as mean $\pm$ standard deviation, while values for Leaves and Fit Time are reported as mean only. For AUC, ACC, and F1, higher is better ($\uparrow$). For Leaves and Fit Time, lower is better ($\downarrow$). The best result in each column for each dataset is bolded. Significant improvements over the best baseline are marked with † ($p < 0.05$).

| Dataset ($N_f, N_s$) | Model | AUC($\uparrow$) | ACC($\uparrow$) | F1($\uparrow$) | Leaves($\downarrow$) | Fit Time (s)($\downarrow$) |
|---|---|---|---|---|---|---|
| Breast-w (9,699) | CART | $0.960 \pm 0.006$ | $0.931 \pm 0.013$ | $0.899 \pm 0.022$ | 11.2 | **0.006** |
| | XGBoost | $0.986 \pm 0.006$ | $0.949 \pm 0.006$ | $0.927 \pm 0.010$ | N/A | 1.733 |
| | Our Method | $\mathbf{0.993 \pm 0.003}$† | $\mathbf{0.956 \pm 0.007}$ | $\mathbf{0.935 \pm 0.010}$ | **2.0** | 0.044 |
| Credit-g (20,1000) | CART | $0.707 \pm 0.019$ | $0.710 \pm 0.017$ | $0.803 \pm 0.013$ | 22.2 | **0.023** |
| | XGBoost | $0.782 \pm 0.007$ | $0.750 \pm 0.005$ | $\mathbf{0.843 \pm 0.003}$ | N/A | 1.155 |
| | Our Method | $\mathbf{0.791 \pm 0.009}$ | $\mathbf{0.760 \pm 0.004}$† | $0.840 \pm 0.005$ | **1.0** | 0.040 |
| Banknote (4,1372) | CART | $0.983 \pm 0.008$ | $0.983 \pm 0.007$ | $0.981 \pm 0.008$ | 19.8 | **0.005** |
| | XGBoost | $0.999 \pm 0.000$ | $\mathbf{0.987 \pm 0.007}$ | $\mathbf{0.986 \pm 0.008}$ | N/A | 1.114 |
| | Our Method | $\mathbf{1.000 \pm 0.000}$ | $0.984 \pm 0.002$ | $0.982 \pm 0.002$ | **2.0** | 0.006 |
| Spambase (57,4601) | CART | $0.924 \pm 0.010$ | $0.900 \pm 0.009$ | $0.868 \pm 0.011$ | 22.2 | **0.030** |
| | XGBoost | $\mathbf{0.981 \pm 0.003}$ | $\mathbf{0.940 \pm 0.008}$ | $\mathbf{0.923 \pm 0.010}$ | N/A | 1.928 |
| | Our Method | $0.966 \pm 0.006$ | $0.922 \pm 0.006$ | $0.900 \pm 0.007$ | **8.0** | 0.901 |
| Diabetes (8,768) | CART | $0.757 \pm 0.020$ | $0.731 \pm 0.010$ | $0.549 \pm 0.089$ | 7.4 | **0.005** |
| | XGBoost | $\mathbf{0.818 \pm 0.014}$ | $0.755 \pm 0.022$ | $0.627 \pm 0.030$ | N/A | 0.306 |
| | Our Method | $0.817 \pm 0.012$ | $\mathbf{0.762 \pm 0.017}$ | $\mathbf{0.629 \pm 0.025}$ | **2.0** | 0.031 |

## L    THE USE OF LARGE LANGUAGE MODELS (LLMS)

Following the ICLR 2026 guidelines regarding the use of Large Language Models (LLMs), we hereby disclose that LLMs were utilized in the preparation of this paper:

- **Text Drafting and Refinement:** Assisting in drafting and revising portions of the text, such as rephrasing existing sentences, checking grammar, and improving linguistic flow.

- **Content Clarification and Summarization:** Helping to clarify complex concepts or summarize lengthy explanations.

---

[1] https://archive.ics.uci.edu/

[2] https://www.cs.toronto.edu/~delve/data/comp-activ/desc.html

[3] https://www.dcc.fc.up.pt/~ltorgo/Regression/DataSets.html

Table 15: Hyperparameters for Classification Datasets

| Dataset | Model | Best Hyperparameters |
|---|---|---|
| Breast-w | CART | {'max_depth': 5, 'min_samples_leaf': 4, 'min_samples_split': 2} |
| | XGBoost | {'learning_rate': 0.05, 'max_depth': 5, 'n_estimators': 200, 'subsample': 0.7} |
| | Our Method | {'max_depth': 1, 'ridge_alpha': 1, 'step_size': 1, 'threshold': 0} |
| Credit-g | CART | {'max_depth': 5, 'min_samples_leaf': 2, 'min_samples_split': 10} |
| | XGBoost | {'learning_rate': 0.01, 'max_depth': 3, 'n_estimators': 200, 'subsample': 0.7} |
| | Our Method | {'max_depth': 1, 'ridge_alpha': 10, 'step_size': 0.01, 'threshold': 0.5} |
| Banknote | CART | {'max_depth': 7, 'min_samples_leaf': 1, 'min_samples_split': 2} |
| | XGBoost | {'learning_rate': 0.1, 'max_depth': 3, 'n_estimators': 200, 'subsample': 1.0} |
| | Our Method | {'max_depth': 1, 'ridge_alpha': 0, 'step_size': 0.01, 'threshold': 0} |
| Spambase | CART | {'max_depth': 5, 'min_samples_leaf': 4, 'min_samples_split': 10} |
| | XGBoost | {'learning_rate': 0.05, 'max_depth': 5, 'n_estimators': 200, 'subsample': 0.7} |
| | Our Method | {'max_depth': 3, 'ridge_alpha': 10, 'step_size': 1, 'threshold': 0} |
| Diabetes | CART | {'max_depth': 3, 'min_samples_leaf': 2, 'min_samples_split': 2} |
| | XGBoost | {'learning_rate': 0.05, 'max_depth': 3, 'n_estimators': 50, 'subsample': 0.7} |
| | Our Method | {'max_depth': 1, 'ridge_alpha': 1, 'step_size': 1, 'threshold': 0} |

- **Formatting and Structural Suggestions:** Providing advice on LaTeX formatting and assisting in structuring certain sections.

All content generated by the LLM underwent rigorous review, editing, and validation by the human authors. The human authors bear full responsibility for the entirety of the paper's content. This disclosure is made to ensure transparency and comply with ICLR's policy.

