# OpenReview forum: "Hinge Regression Tree: A Newton Method for Oblique Regression Tree Splitting"
_ICLR.cc/2026/Conference — ICLR 2026 Poster_

### Official Review · Reviewer_AC5G · 2025-10-31

**Soundness:** 3
**Presentation:** 3
**Contribution:** 3
**Rating:** 6
**Confidence:** 2

**Summary:**

The paper introduces hinge regression tree (HRT), a new method for learning oblique regression tree splits. The key idea is to formulate each split as a nonlinear least-squares problem over two linear predictors, yielding an alternating fitting procedure that is mathematically equivalent to a damped Newton method within fixed partitions. HRT demonstrates competitive performance in benchmarking experiments compared to standard baselines. Overall, the proposed method appears promising. My comments are as follows:

- Section 3: Consider adding a short subsection before Section 3.1 that briefly recaps oblique decision tree methods. The current text is related but does not explicitly state that this approach is an instance of oblique decision tree regression.

- Line 154: I believe this represents a linear decision boundary, rather than a hinge-based one.

- Line 191: The optimization behavior accounting for partition changes is not analyzed, leaving a disconnect between the theoretical guarantee and the practical performance. The empirical results show that the algorithm converges on real data, which is reassuring; however, I would still view the algorithm as heuristic, and suggest avoiding terms such as “rigorously” (line 49) or “solid theoretical foundation” (line 50).

- The computational efficiency of the proposed algorithm has not been comprehensively evaluated.

**Strengths:**

See summary above.

**Weaknesses:**

See summary above.

**Questions:**

See summary above.

---

> ### Author Response · Authors · 2025-11-22
>
> Thank you for your valuable comments. We have refined the definitions and added efficiency analysis.
>
>
> **1. Section 3 Structure & Global Objective (Section 3):**
> We followed your suggestion to clarify the method's positioning.
> *   **Recap:** We have revised the beginning of **Section 3** to explicitly recap oblique decision tree methods and state that HRT is a specific instance of this framework.
> *   **Global Objective:** In **Section 3.1**, we formally defined the global objective function (Eq. 1) to provide clear context. This sets the stage for the subsequent node-level optimization, showing exactly how the local splits contribute to minimizing the global regression error.
>
> **2. Linear vs. Hinge Boundary:**
> We clarified the terminology in Section 3.4. The decision boundary $\tilde{x}^T(\theta_{t_1} - \theta_{t_2}) = 0$ is a linear hyperplane.
>
>
> **3. Optimization & Partition Changes:**
> We have clarified the connection between theory and practice by explicitly addressing partition changes.
>
> * **Update:** In **Appendix B**, we now provide an analysis that incorporates the effect of partition updates. We show that a **backtracking line search** guarantees a **monotonic decrease** of the node-level objective even when partition boundaries shift, which fills the gap noted by the reviewer and explains the observed stability of the algorithm in practice.
>
> In line with this clarification, we have also **removed phrases such as “rigorously” and “solid theoretical foundation”** from the introduction and adjusted the wording to more modestly describe our theoretical guarantees.
>
>
> **4. Computational Efficiency:**
> We added **Table 4** to report training times. HRT remains highly efficient (seconds to minutes) even on larger datasets.

---

### Official Review · Reviewer_e2Hz · 2025-11-01

**Soundness:** 2
**Presentation:** 2
**Contribution:** 2
**Rating:** 2
**Confidence:** 4

**Summary:**

The paper proposes the Hinge Regression Tree (HRT), an algorithm for building oblique regression trees by jointly learning two linear predictors at each split. Each node models its output as a hinge function, leading to a piecewise linear regression surface. Training alternates between (i) fitting the two linear models with ordinary least squares (optionally ridge-regularized) and (ii) reassigning data to the branch that yields the higher prediction. The authors describe this alternating fitting as a damped Newton method within fixed partitions and show that, with small damping factors, it converges stably in practice. They also restate a known universal-approximation result for piecewise linear models and present experiments on synthetic and real regression datasets, comparing HRT to CART, TAO, DGT, DTSemNet, and XGBoost.

**Strengths:**

1. The node training procedure (alternate OLS fits for two linear predictors under a hinge) is simple, transparent, and easy to re-implement. The paper provides explicit pseudocode and a complete build procedure, which supports reproducibility.
2. Compared to the baseline methods, HRT enjoys competitive performance.
3. The paper is mostly clear and easy to understand.

**Weaknesses:**

1. The contribution is overstated. Within fixed partitions, the update equals a Gauss–Newton/OLS step (standard), which is not difficult to prove. However, the hard part in the decision tree method is partition switching. Unfortunately, there is no guarantee of monotone decrease or convergence of the alternating fit-reassign procedure. I am not convinced that this part should be left to future work, as this guarantee is far more interesting and important than the updates in each subspace to show the validity of the proposed method.

2. The discussion of the damping effect is insufficient. For example, the results improvement for $\mu=0.01$ (damping) and $\mu=1.00$ (no damping) is only marginal in Tables 4 and 5. If using $\mu=1.00$ (OLS) and the standard fallback algorithm can provide competitive results, then the advantage of the damping term, which is a contribution of the paper, is unclear, and the reformulation into Gauss-Newton update seems to be unnecessary.

3. More benchmarks could be given to further show the effectiveness of the methods and examine the alternating fit-reassign procedure. In particular, high-dimensional datasets such as Communities & Crime (UCI), BlogFeedback (UCI), and baselines such as LightGBM can be included.

4. Some of the improvements of HRT over other methods are marginal (within 1%). Statistical significance tests should be given to potentially make many differences statistically distinguishable.

5. Although the authors discussed the complexity, there is no comparison of the computational time, especially how much time is sacrificed for using damped optimization. This makes the efficiency of the proposed method in practice hard to assess.

6. The technical contribution of the theory is limited. The approximation of piecewise linear functions to any $C^2$ target in many similar cases is known, e.g., Breiman (1993), and ReLU approximation Barron (1993). The sample complexity of Oblique trees is also established in Cattaneo et al. (2024). Therefore, the approximation result is not surprising.

Barron (1993): Universal approximation bounds for superpositions of a sigmoidal function, IEEE Transactions on Information Theory

**Questions:**

See weakness.

---

> ### Author Response · Authors · 2025-11-22
>
> Thank you for your detailed feedback. We have filled the theoretical gaps and expanded benchmarks.
>
> **1. Convergence of Partition Switching:**
> We agree that partition switching is the core challenge. In **Appendix B**, we prove that by using a **backtracking line-search**, we can find a step size $\mu$ such that the node-level objective strictly decreases (or stabilizes). This theoretically handles the discrete nature of partition switching and guarantees a monotone decrease of the node-level objective.
>
> **2. Efficacy of Damping & Time Cost:**
> We conducted a detailed ablation study in **Table 7 (Appendix H)** on the *Kinematics* dataset, comparing the unit step ($\mu=1.0$) vs. our adaptive damping ($\mu=\text{auto}$).
> *   **Result:** Damping actually **saves** time.
>     *   **$\mu=1.0$ (No damping):** The algorithm is unstable and oscillates, requiring **83.43 iterations** and **1.88s** to converge (RMSE 0.107).
>     *   **$\mu=\text{auto}$ :** The algorithm converges efficiently in **15.76 iterations** and only **0.79s** (RMSE 0.102).
>     *   Damping does not compromise time; instead, it prevents oscillations and partition collapse.
>
> **3. Benchmarks:**
> *   **New Data:** We added the requested **Communities & Crime** and **BlogFeedback** datasets to Table 2. HRT performs competitively on both.
> *   **Linear Baselines:** We added **Linear Tree** and **M5** to show HRT's strength within the "linear leaf" model class.
> *   **LightGBM:** We focused our comparison on **single-tree** models. We already included **XGBoost** (Table 2) as a representative of ensemble methods. Comparing a single estimator (HRT) against multiple boosting ensembles (like LightGBM) would be an unfair comparison of model capacity, though we note HRT remains competitive with XGBoost on several tasks.
> *   **Significance:** We applied statistical tests (t-test) and marked significant wins with $\dagger$ in Tables 1 & 2.
>
>
> **4. Training Time:**
> As shown in **Table 4**, HRT is highly efficient (seconds to minutes) and competitive with other decision tree methods, significantly outperforming differentiable tree baselines like DGT in speed.

---

> > ### Comment · Reviewer_e2Hz · 2025-11-27
> >
> > I appreciate the author's great effort in addressing my concerns. The additional experiments for the runtime and damping term showed the advantage of the proposed method. However, I am still not convinced that Theorem 2 correctly handles the situation of partition switching. In particular, (a) is based on a fixed partition, deduced from equations (7-8). (b) is an interesting result leveraging the "discrete" nature of the dataset, but seemingly it asserts that there exists a $\mu$ to make the objective function decrease while preserving the partition structure (line 819-820). So it can not give a bound for the partition switching case. I wonder if the author can verify this. Given the current result, I have increased my score to 4.

---

> > > ### Author Response · Authors · 2025-11-28
> > > **Clarification on Theorem 2 and Partition Switching**
> > >
> > > Dear Reviewer e2Hz,
> > >
> > > Thank you for your insightful feedback and for increasing the score. We would like to address your concern about partition switching in Theorem 2 with a clearer explanation.
> > >
> > > You are absolutely correct that the proof of the **existence** of a descent step (Step 2 in Appendix B, lines 819-820) relies on the fixed-partition property. This guarantees that, for a sufficiently small $\mu$, the partition structure remains locally invariant. **However, this does not limit the algorithm to fixed partitions.** The logic is as follows:
> > >
> > > 1. **Guarantee of Non-Stagnation (The "Bound"):**
> > >    The fixed-partition property serves as a **safety guarantee**. It proves that a valid descent step *always exists*. In the worst case, we can find a small enough $\mu$ that keeps the algorithm within the current smooth region and ensures a decrease in the objective $V$. This guarantees that the algorithm will never get stuck at a non-stationary point.
> > >
> > > 2. **Handling Partition Switching via Line Search:**
> > >    In practice, the backtracking line search allows larger steps that **switch partitions**, as long as the descent condition $V(\theta^{(k+1)}) < V(\theta^{(k)})$ is satisfied. This ensures that even when partition switching occurs, the step is accepted if it leads to a decrease in the objective.
> > >
> > > 3. **Global Convergence:**
> > >    Since a descent step is always guaranteed to exist (point 1) and any accepted step (whether it switches partitions or not) strictly reduces the objective (point 2), the sequence $\{V(\theta^{(k)})\}$ is strictly decreasing and bounded below by 0. By the Monotone Convergence Theorem, the objective value must converge globally.
> > >
> > > **In summary:**
> > > The fixed partition property provides the theoretical guarantee for the *existence* of a valid descent step, while the line search mechanism allows *partition-switching* steps that accelerate convergence, provided they satisfy the descent condition.
> > >
> > > We hope this clarifies that partition switching is implicitly handled: if it results in a valid decrease, the step is accepted; if not, the step size adjusts to ensure smooth progression.

---

### Official Review · Reviewer_GFxH · 2025-11-02

**Soundness:** 1
**Presentation:** 2
**Contribution:** 1
**Rating:** 2
**Confidence:** 5

**Summary:**

The paper proposes to learn an oblique decision tree with linear leaf predictors using the traditional greedy recursive partitioning approach but with the variation on the splitting procedure. The splitting procedure uses the hyperplanes at both leaves to define the splitting hyperplane: $\mathbf{\theta_1} - \mathbf{\theta_2}$. The learning algorithm uses the current split to define Newton updates, but it is not clear how it relates to finding the global optimum. Experiments are performed on synthetic and real datasets showing improved accuracy of the proposed method.

**Strengths:**

Oblique decision trees are an important model class which has not been as widely studied. This paper proposes one approach in learning this relatively unexplored model class.

**Weaknesses:**

* Objective function is defined only for a splitting criterion. It is not clear what is the global objective being optimized.

* During splitting, the internal decision node hyperplane is defined by its two linear leaf weights. Ideally, these sets of parameters (decision split hyperplane and its two children linear weights) must be independent. It is not clear why this way of coupling is used.

* It is not clear whether the proposed algorithm optimizes eq. 1. No theoretical guarantees of convergence or optimality is shown.

* The comparison experiments on real world data are not apples-to-apples. The baselines of CART and TAO, for example, use constant leaves, while the proposed approach uses linear leaves. Piecewise constant models are in general not suitable for regression problems.

**Questions:**

No questions.

---

> ### Author Response · Authors · 2025-11-22
>
> Thank you for your review. We have addressed your concerns regarding the objective function and baselines.
>
> **1. Global Objective & Optimization:**
> You raised concerns about the clarity of the global objective and whether the algorithm effectively optimizes it. We have clarified this:
> *   **Definition:** As explicitly defined in **Section 3.1 (Eq. 1)**, our global objective is the sum of squared errors across all leaf nodes. Like standard decision trees (e.g., CART), HRT optimizes this global objective via a greedy recursive partitioning strategy.
>
>
> **2. Coupling of Hyperplane and Weights:**
> You asked why the split hyperplane depends on leaf weights ($\tilde{x}^T(\theta_1 - \theta_2) = 0$).
> *   **Explanation:** This is intrinsic to the **Hinge function** formulation $h(x) = \max(\ell_1(x), \ell_2(x))$. The geometry implies that the "switch" between predictors happens exactly where they intersect.
> *   **Benefit:** This design reduces the parameter count (no separate gating vector needed) and provides a highly expressive **piecewise linear** model locally, similar to canonical Representation for piecewise linear functions (Chua & Kang, 1977), without the need for arbitrary separation parameters.
>
> **3. Node-level convergence:**
> * **Node-level optimization guarantees.** To clarify how the algorithm optimizes the node-level objective (**Eq. 2 in the revised paper, originally Eq. 1**), we have added Appendix B, where we prove that our alternating fit–reassign scheme with backtracking line search guarantees a monotone decrease of the node-level objective, and that, once the partition stabilizes, the Newton iterates converge to the unique OLS minimizer for that fixed partition.
>
>
> **4. Apples-to-Apples Comparison:**
> To address your concern about comparing linear leaves (HRT) vs. constant leaves (CART), we added two linear-leaf baselines: **M5 Model Tree** (Wang & Witten, 1997) and **Linear Tree** (Cerliani, 2022).
> *   **Result:** As shown in **Table 2**, HRT outperforms both. For instance, on *Kinematics*, HRT achieves significantly lower RMSE (0.102) compared to M5 (0.175) and Linear Tree (0.141), validating the superiority of our hinge-based splitting optimization over standard linear trees.

---

### Official Review · Reviewer_GmnV · 2025-11-02

**Soundness:** 3
**Presentation:** 3
**Contribution:** 3
**Rating:** 6
**Confidence:** 4

**Summary:**

This paper proposes a novel splitting method for oblique regression trees. It uses two linear predictors and the splitting depends on which predictor is larger or smaller. This splitting problem is solved via an alternating fitting procedure which is equivalent to a damped Newton method within fixed partitions. This paper proves that the oblique regression tree with such a splitting mechanism is a universal approximator.

**Strengths:**

1. This paper is well written with clear content organization, method description, and mathematical notation.
2. The splitting mechanism of oblique regression tree is novel, i.e. the combination of two linear predictors and hinge function.
3. This paper proves that the proposed method is a piece-wise linear model class which is a universal approximator. Thus its expressive power is underpinned by theoretical foundation as well as experimental results.

**Weaknesses:**

1. Some implementation details are missing, such as how to initialize two linear predictors.
2. The formal global convergence proof is not provided. I understand this is challenging. But I think the authors can try some weaker conclusions. For example, under what conditions is the loss function monotonically decreasing?
3. It is better to add experiments for binary classification tasks.

**Questions:**

On average, how many iterations are needed to split a node?

---

> ### Author Response · Authors · 2025-11-22
>
> We appreciate your positive assessment. We have addressed your questions with specific implementation details and new experiments.
>
> **1. Implementation Details: Initialization:**
> You correctly pointed out the need for initialization details. We have added **Appendix D.1** to explicitly describe our robust initialization strategy, which consists of three steps to ensure stability:
> *   **Step 1 (Coarse Split):** We temporarily split the data based on the median of the feature with the maximum variance.
> *   **Step 2 (Ridge Initialization):** We fit two initial OLS (Ridge) Regression models on these coarse subsets to obtain $\theta_1^{(0)}$ and $\theta_2^{(0)}$.
> *   **Step 3 (Perturbation Safeguard):** If the coarse split yields identical parameters or fails, we fallback to initializing both $\theta$ near the global OLS solution with small random perturbations to ensure $\theta_1 \neq \theta_2$ (breaking symmetry) and kickstart the optimization.
>
> **2. Formal Convergence Proof:**
> Following your suggestion, we have proved in **Appendix B** that with a backtracking line-search mechanism, the node-level objective (Eq. 2 in the revised paper) is **monotonically decreasing** throughout the iterations, leading to convergence.
>
> **3. Binary Classification:**
> We extended HRT to binary classification. As detailed in **Appendix L**, HRT achieves competitive AUC/F1 scores while maintaining significantly shallower trees compared to baselines.
>
> **4. Iterations to Split:**
> We added **Table 4** showing the average iterations. Typically, it takes **~6.7 iterations** for easy splits (e.g., Delta Ailerons dataset) and **~45.7 iterations** for harder tasks (e.g., Fried), demonstrating the algorithm's adaptivity.

---

### Author Response · Authors · 2025-11-23
**General Response to All Reviewers**

We thank the reviewers for their constructive feedback and insightful comments. We have revised the manuscript to address the concerns regarding theoretical convergence, baseline comparisons, and implementation details. All major revisions are marked in **blue** in the updated paper. Below is a summary of the key improvements:

### 1. Theoretical Convergence Analysis (Addressing Reviewers GmnV, GFxH, e2Hz, AC5G)
A shared concern among reviewers was the theoretical guarantee of the alternating optimization procedure, specifically regarding the discrete nature of partition switching.
*   **Monotonicity Proof (Appendix B):** We have added a formal proof demonstrating that by incorporating a **backtracking line-search strategy**, the node-level objective function (Eq. 2) decreases monotonically throughout the iterations.
*   **Convergence:** We explicitly show that once the partition stabilizes, the iterative updates converge to the unique OLS minimizer for the corresponding subspace. This provides the missing theoretical link between the partition assignment and the Newton update.

### 2. "Apples-to-Apples" Baselines & New Benchmarks (Addressing Reviewers GFxH, e2Hz)
To address concerns that comparing our linear-leaf model (HRT) against constant-leaf models (e.g., CART) was unfair, and to test on higher-dimensional data:
*   **Linear-Leaf Baselines:** We added **M5 Model Tree** (Wang & Witten, 1997) and **Linear Tree** (Cerliani, 2022) to **Table 2**. HRT consistently outperforms these linear-leaf baselines (e.g., on Kinematics, HRT RMSE **0.102** vs. M5 **(0.175)** and Linear Tree **(0.141)**), validating the specific effectiveness of our hinge-based splitting optimization.
*   **New Datasets:** We included high-dimensional datasets (**Communities & Crime**, **BlogFeedback**) as requested.
*   **Significance Tests:** We applied t-tests to our results, marking significant improvements with $\dagger$ in Tables 1 & 2.

### 3. Efficiency and Damping Ablation (Addressing Reviewers e2Hz, AC5G)
We investigated the necessity of the damped Newton step and evaluated computational costs.
*   **Damping Improves Speed (Appendix H):** We added an ablation study (**Table 7**) showing that the damped step ($\mu < 1$) is critical for efficiency, not just stability. Using a unit step ($\mu = 1$) causes oscillation and slower convergence (1.88s), whereas adaptive damping converges significantly faster (0.79s).
*   **Training Time:** We added **Table 4** to report average training times and iteration counts. HRT proves to be highly efficient (seconds to minutes), outperforming differentiable tree baselines like DGT.


### 4. Binary Classification and Initialization (Addressing Reviewer GmnV)
*   **Binary Classification (Appendix L):** We successfully extended HRT to binary classification. The results show competitive AUC and F1 scores with significantly shallower tree depths compared to baselines.
*   **Initialization Details (Appendix D.1):** We have explicitly described the initialization strategy (Coarse Split $\to$ Ridge Initialization $\to$ Perturbation Safeguard) to ensure the method is fully specified.

### 5. Clarification on Global Objective (Addressing Reviewers GFxH, AC5G)
*   **Global Objective Definition:** We clarified in **Section 3.1** (Eq. 1) that HRT optimizes the global regression error via recursive partitioning. This resolves the confusion regarding the relationship between the node-level splitting criterion and the global tree construction objective.

---

### Author Response · Authors · 2025-11-30
**Author Summary for the New AC after Review Freeze**

Dear Area Chair,

We are grateful for your efforts in reassessing our submission. Below we briefly summarize our paper and how the **revised version (substantially changed in both theory and implementation)** addresses the main points raised in the initial reviews.

---

### 1. Paper in one sentence

We propose the **Hinge Regression Tree (HRT)**, an oblique regression tree where each split is a hinge over two linear predictors, trained via an alternating fit–reassign procedure that corresponds to a damped Newton (Gauss–Newton) method within fixed partitions; in the revised version we equip it with a backtracking line search and prove monotone decrease of the node-level objective. We demonstrate strong empirical performance on both synthetic and real-world datasets.

---

### 2. Convergence, partition switching, and the new line search

Several reviewers (GmnV, GFxH, e2Hz, AC5G) asked:

> When partitions can change, does the alternating fit–reassign procedure have any guarantee of convergence or monotone decrease?

In the **original submission**, we only showed that within a fixed partition the update reduces to a Gauss–Newton/OLS step and did **not** address the case where partitions may switch.

In the **revised paper**, we introduce a **backtracking line search** at the node level and base our analysis on this variant (Appendix B, Theorem 2). We show that on regions where the partition is fixed, the Newton direction is a strict descent direction and a sufficiently small step always decreases the node-level objective. The line search may accept larger steps that change the partition as long as they satisfy the descent condition; otherwise, the step size is reduced until a descent step is found. Consequently, every **accepted** step strictly decreases the node-level objective, which is bounded below by 0, so the objective values form a **monotonically decreasing and convergent sequence**. Once the partition stabilizes, the iterates converge to the OLS minimizer for that fixed partition. This line-search variant is new in the revised version and directly addresses convergence-related concerns about partition switching, monotone decrease, and potential oscillations.

---

### 3. Other major concerns

**(A) “Apples-to-apples” baselines and high-dimensional datasets**
We added **two linear-leaf baselines** in revised **Table 2**—**M5 Model Tree** and **Linear Tree**—in addition to constant-leaf trees (CART, TAO). HRT outperforms these linear-leaf baselines (e.g., on **Kinematics**, RMSE 0.102 vs. 0.175 and 0.141), indicating that gains come from our hinge-based splitting rather than just using linear leaves. We also added **higher-dimensional datasets** such as Communities & Crime and BlogFeedback, where HRT remains competitive, and we now report **t-tests**, marking statistically significant improvements in Tables 1–2.

**(B) Damping, efficiency, and runtime**
In revised **Appendix H**, we perform an ablation on **step size / damping**, comparing unit-step updates (μ = 1.0, no damping) with adaptive damping using backtracking line search. Unit-step updates tend to oscillate, require more iterations, and lead to longer training times, whereas adaptive damping is more stable and typically faster in wall-clock time, with comparable or better RMSE. A new **training time table** reports average wall-clock time and iteration counts; HRT typically trains in **seconds to minutes**, is noticeably faster than differentiable tree baselines such as DGT, and remains competitive with XGBoost in accuracy.

**(C) Implementation details and binary classification**
We added a detailed **initialization strategy** in revised **Appendix D.1** (coarse split on the feature with maximum variance, ridge initialization on coarse subsets, and perturbation-based safeguards), making the method fully reproducible from the pseudocode. We also extended HRT to **binary classification** in **Appendix L**, reporting AUC/F1 scores; HRT achieves competitive performance with significantly shallower trees compared to baselines, suggesting that the hinge-based splitting mechanism transfers naturally to classification.

**(D) Global objective and positioning among oblique trees**
In **Section 3.1 (Eq. (1))**, we now clearly define the **global objective** as the sum of squared errors over all leaves, optimized via standard greedy recursive partitioning in direct analogy to CART. At the beginning of Section 3, we explicitly position HRT as an instance of **oblique regression trees** within this framework. The coupling between the split hyperplane and the two linear predictors is induced by the hinge construction: the decision boundary is where the two predictors intersect, which reduces the number of free parameters (no separate gating vector) while retaining the expressive power of piecewise linear canonical forms.

---

Thank you for your time and for considering our work under these unusual circumstances.

---

### Meta-Review · Area_Chair_8NFG · 2026-01-06

**Summary:**

The paper presents a novel greedy method for learning oblique regression tree (that can be easily extended to binary classification) with hinge function splittings.
In the final version of the paper, node-level convergence (non trivial result) is proved, and other properties are shown: universal approximation, equivalence of node-level fitting to Gauss-Newton method with fixed data partitions.
The resulting model is competitive in particular with existing and recent oblique tree methods, and efficient even on high dimensional datasets.
The reviewers' feedback significantly helped improving the work, which is now suitable for acceptance.

**Reviewer Concerns:**

Solved issues include:
1. **Lack of non-trivial theoretical results and lack of convergence proofs.** Node-level convergence is now proved.
2. **Missing high dimensional datasets and oblique tree baselines.**
3. **Lack of significance tests.** The main tables now indicate which results are significantly better than the baselines' ones, showing that the method is significantly better on 3 out of 14 datasets, and has similar performance to the best baseline on 6.
4. **Missing training times.** Table 4 shows the competitiveness of the proposed method w.r.t. related baselines.

Outstanding issues:
1. During the rebuttal, the authors clarified the initialization needed to stabilizing training. An ablation is missing to assess the empirical impact of such procedure.

**Reviewer Scores:**

GmnV: The score should have increased given the clarifications, new convergence result and new experiments on binary classification.
AC5G: They might have kept their score to 6, given that the raised issues were not major.
e2Hz: They stated their score was increased. The major issues were solved, so their score should have changed to a 6.
GFxH: Their score should have increased at least to a 4. Major issues were addressed. It is unclear whether the justification for using the hinge function was satisfactory.

---

### Decision · Program_Chairs · 2026-01-26

Accept (Poster)